

# Comparison of CryoSat-2 and ENVISAT freeboard height over Arctic sea ice: Toward an improved Envisat freeboard height retrieval.

Kevin Guerreiro[1], Sara Fleury[1], Elena Zakharova[1,2], Alexei Kouraev[1,2,3], Frédérique Rémy[1], and Philippe Maisongrande[1]

[1]Laboratoire d'Etudes en Géophysique et Océanographie Spatiales, Centre National de la Recherche Scientifique (LEGOS - CNRS, UMR5566), Université de Toulouse, 31400 Toulouse, France
[2]State Oceanography Institute, St. Petersburg Branch, St. Petersburg, Russia
[3]Tomsk State University, Tomsk, Russia

*Correspondence to:* Guerreiro Kevin, kevin.guerreiro@legos.obs-mip.fr

**Abstract.** During the past decade, sea ice freeboard height has been monitored with various satellite altimetric missions with the aim of producing long-term time series of ice thickness. To achieve this goal, it is essential to analyze potential inter-mission biases and to produce freeboard height datasets as free of instrumental error as possible. In the present study, we compare Envisat and CryoSat-2 freeboard height during the common flight period (2010-2012). Our results show that Envisat

5   freeboard height is always thinner (-14 cm in average) when compared to CryoSat-2 (3 cm in average). In addition, Envisat freeboard height displays an unrealistic negative growth from November to April (-2.4 to -3.7 cm) while CryoSat-2 dispalys a positive and coherent winter growth (2.4 to 2.7 cm). The discrepancy between the two datasets is found to be related to a dissimilar impact of ice roughness and snow volume scattering on SAR (CryoSat-2) and pulse-limited (Envisat) altimetry. Following this result, we show that the freeboard height difference between the two datasets can be expressed as a function of

10  the waveform pulse-peakiness. Based on the relation between the Envisat pulse-peakiness and the freeboard height difference, we produce a monthly CryoSat-2-like version of Envisat freeboard height that reduces the average RMSD with CryoSat-2 from ∼16 cm to ∼2 cm and improves the freeboard height growth cycle (2 - 3 cm). The comparison of the altimetric datasets with *in situ* ice draft measurements during the common flight period shows that the corrected Envisat dataset (RMSE = 16 - 29 cm) is as accurate as CryoSat-2 (RMSE = 13 - 25 cm) and highly more accurate than the uncorrected Envisat dataset (RMSE =

15  108 - 132 cm). The comparison of the improved Envisat freeboard height dataset is then extended to the rest of the Envisat mission to demonstrate the validity of the improved Envisat dataset out of the calibration period. As a result, we find a good agreement between the Envisat and the *in situ* ice draft datasets (RMSE = 14 - 30 cm), which demonstrates the potential of the pulse-peakiness-correction to produce accurate freeboard height estimates over the entire Envisat mission. Finally, we show the averaged-circumpolar ice thickness variations from 2002 to 2015 by combining CryoSat-2 and Envisat datasets.



# 1 Introduction

Sea ice is one of the most sensitive indicators of the Arctic climate system changes. Satellite observations have demonstrated that Arctic sea ice extent has decreased at an average rate of 4% per decade from 1978 to 2010 and at an accelerated rate of 8.3% during the 1996-2010 period (Comiso, 2012). In addition to the sea ice cover reduction, it has been shown that Arctic sea ice is

also thinning (Rothrock et al., 1999; Kwok and Rothrock, 2009). Based on submarine ice draft measurements, Rothrock et al. (1999) reported a decrease of ∼1.3 m in the 1990s relatively to ice thickness measurements obtained during the 1958-76 period. However, Holloway and Sou (2002) show that local submarine measurements can be impacted by large scale displacement of perennial ice and that sea ice thickness should be monitored at a basin-scale to accurately estimate sea ice volume changes.

Over the past decade, satellite radar altimeters have been used to estimate basin-scale sea ice thickness. It is generally

assumed that over snow-covered sea ice, the main scattering horizon of the Ku-band (13.6 GHz) radar signal is located at the snow/ice interface (Beaven, 1995). Hence, Ku-band radar altimeters have been used to monitor the height of sea ice above sea level, generally referenced as sea ice freeboard height. By assuming the hydrostatic equilibrium between the ocean and the snow-covered sea ice, freeboard height can be converted to ice thickness (Laxon et al., 2003). Following this methodology, sea ice thickness has been estimated with the Low Resolution Mode (LRM) altimeters RA onboard ERS-1 and RA-2 onboard

ERS-2 and Envisat (Laxon et al., 2003; Giles et al., 2008). While the Envisat ice thickness estimates are promising, the current prototype product (http://icdc.cen.uni-hamburg.de/1/projekte/esa-cci-sea-ice-ecv0.html) has a positive bias of 0.5 to 1.5 m and does not reproduce accurately the seasonal cycle of ice growth. It is the purpose of the European Space Agency (ESA) Sea Ice Climate Change Initiative project (SI-CCI) to improve RA-2 freeboard height retrievals and to provide accurate time series of ice thickness over the Envisat mission lifetime (Ridout and Tonboe., 2012). More recently, sea ice thickness was estimated

with the Ku-band Synthetic Aperture Radar (SAR) Interferometric Radar Altimeter (SIRAL) onboard CryoSat-2. Unlike for Envisat, CryoSat-2 ice thickness estimates are in good agreement with *in situ* measurements and display a realistic seasonal cycle (Laxon et al., 2013; Kwok and Cunningham, 2015).

The higher accuracy of Cryosat-2 ice thickness estimates as compared to Envisat ones rise an important question: Why does CryoSat-2 provide better estimates of ice thickness than Envisat while both sensors operate at the same central frequency?

First of all, the bias in Envisat ice thickness estimates could be caused by an inaccurate conversion of freeboard height to ice thickness. Indeed, ice thickness is generally obtained by resolving the hydrostatic balance between the ice and the ocean. As there is yet, no collocated measurements of ice and snow properties, empirical parametrization of snow depth and ice density are generally used to derive ice thickness from freeboard height. While this empirical parametrization represents so far the only way to convert freeboard height to ice thickness, it is an important source of uncertainties. Having said that, Kern et al.

(2015) use a similar parametrization of snow depth and ice density as in CryoSat-2 studies (Laxon et al., 2013; Kwok and Cunningham, 2015) to convert Envisat freeboard height to ice thickness. As a result, they find that the accuracy of Envisat ice thickness estimates is lower than CryoSat-2's, despite the use of a similar parametrization. In particular, they do not succeed to reproduce accurately the seasonal cycle of ice growth. This result suggests therefore that the bias in Envisat ice thickness



estimates is driven by a bias in the freeboard height fields rather than by an inaccurate conversion of freeboard height to ice thickness.

To analyze the potential bias in the Envisat freeboard height fields, the common flight period of Envisat and CryoSat-2 (November 2010-March 2012) represents an unique opportunity. In Schwegmann et al. (2015), Envisat and CryoSat-2 free-board height datasets are compared over Antarctic sea ice. While the spatial and temporal distributions are consistent, it is shown that CryoSat-2 freeboard height is thicker (thinner) for thick (thin) freeboard values. The authors conclude that the discrepancy between the two datasets is likely related with the difference of characteristics between LRM and SAR altimetry: Over sea ice, conventional LRM altimeters such as Envisat have a typical footprint of 2-10 km (Connor et al., 2009). As the LRM technology does not allow to localize the origin of each scatterers within this large ground footprint, bright off-nadir reflexions can drive important biases on the monitoring of surface level position. Unlike LRM altimetry, the SAR technology and the Doppler post processing allow to identify the along track position of each scatters and offer SIRAL an along track footprint of $\sim 300$ m (the across track being unchanged), which nearly eliminate front and back off-nadir reflections and also significantly reduced the impact of lateral off-nadir reflection. This new footprint geometry leads to sharpen the response func-tion of the radar signal relatively to a LRM radar altimeter (Raney, 1998). This improvement of the SAR technology limits the impact of ice surface diffusion (ice roughness and snow volume scattering) on the radar waveform echoes, which should allow to improve the surface level estimates accuracy (Raney, 1995). Hence, the more accurate freeboard estimates obtained with CryoSat-2 are likely related to the reduced impact of ice surface diffusion on SAR altimetry. Following this theoretical basis, we seek to analyze the discrepancy of freeboard height between CryoSat-2 and Envisat and its link with the ice surface diffusion variability.

In section 2, we introduce the datasets used in the present study as well as the freeboard height retrieval algorithms. Then, we compare the spatial and seasonal variability of Envisat and CryoSat-2 freeboard height fields (section 3.1). In section 3.2 we show the link between the freeboard height discrepancy and the ice surface diffusion. In section 3.3, we build a Cryosat-2-like version of Envisat freeboard height based on the relation between the freeboard height discrepancy between Envisat and CryoSat-2 and the Envisat pulse-peakiness. In section 3.4, we convert CryoSat-2, Envisat and the corrected Envisat freeboard height datasets to ice draft fields and compare each dataset to *in situ* measurements. Finally, we estimate pan-Arctic seasonal ice thickness over the 2002-2015 period by combining Envisat and CryoSat-2 datasets (section 3.5).

## 2 Data and methodology

### 2.1 Envisat

Envisat was launched in 2002 by ESA and was set on the same orbit than the ERS-1/2 satellites, providing coverage of the Arctic Ocean up to 81.5°N. The RA-2 altimeter onboard Envisat includes a Ku-band pulse-limited radar altimeter with a bandwidth of 320 MHz. To derive the Envisat freeboard height fields, we use the Sensor Geophysical Data Record (SGDR) product from ESA (http://www.aviso.altimetry.fr/) that are converted to NetCDF files by the Center of Topography of Oceans and Hydrosphere (CTOH) with recent corrections of mean sea surface height DTU15, (Andersen and Knudsen, 2015) and tide




corrections FES14, (Carrere et al., 2015). Envisat freeboard height is estimated from November 2010 to April 2011 for the first common winter and from November 2012 to March 2012 for the second common winter season. The reduced period during winter 2011/2012 is due to the end of the Envisat mission at the beginning of April 2012.

## 2.2 CryoSat-2

CryoSat-2 was launched by ESA in 2010 and was primarily designed for the observation of ice over land and ocean surfaces (Wingham et al., 2006). Although the highly inclined orbit of CryoSat-2 allows the monitoring of sea ice freeboard height up to 88°N, we only make use of observations bellow 81.5°N in the present study (maximum latitude of Envisat orbit). Bellow this latitude and over the Arctic ocean, CryoSat-2 operates mostly in SAR mode except for observations close to the coastline and for a narrow patch located between 130°W and 150°W and at north of 80°N where it is set in SAR Interferometry mode.

As the purpose of the present study is to analyze the difference of freeboard height between LRM and SAR altimetry, we only consider freeboard height observations in SAR regions and discard observations in SARIn regions for both CryoSat-2 and Envisat. The SIRAL altimeter onboard CryoSat-2 operates at the same central frequency (Ku-band) than Envisat with a similar bandwidth (320 MHz) as Envisat. In the present study, we produce freeboard height fields from the CryoSat-2 Baseline-C l1b product that is also converted to NetCDF files by the CTOH with the same sea level corrections as for Envisat. As for Envisat,

CryoSat-2 freeboard height is estimated from November 2010 to April 2011 for the first common winter and from November 2012 to March 2012 for the second common winter season.

## 2.3 Freeboard retrievals

The procedure we apply to retrieve freeboard height from Envisat and CryoSat-2 is described in details in the ESA SI-CCI project and is based on the already published method used for ERS-2 (Peacock and Laxon, 2004; Laxon et al., 2003). Freeboard

height is obtained by measuring the difference of range between the ocean and the sea ice floes. The discrimination between sea ice and ocean observations is performed through the analysis of the radar waveform shape. Indeed, radar echoes over open water in sea ice fractures (leads) are generally highly specular due to the presence of a thin and smooth layer of ice and/or to a flat ocean surface. On the opposite, the radar return over sea ice floes is relatively diffuse due to the impact of sea ice deformation and snow accumulation. Consequently, the discrimination between leads and ice floes can be performed with the

analysis of the pulse peakiness (PP) (Peacock and Laxon, 2004). In the present study, the PP is defined as follows :

$$PP = \frac{max(WF)}{\sum_{i=0}^{i=N_{WF}}(WF_i)} \qquad (1)$$

Where WF represents the waveform echoes and $N_{WF}$ is the number of range bins. In most studies, the upper limit of PP used to identify sea ice floes is different from the lower limit used to identify leads (Kurtz et al., 2014; Ricker et al., 2014; Laxon et al., 2013). Hence, a large number of observations are neither considered as leads nor as ice floes. These observations,

characterized by an intermediate value of PP, arise most likely from thin and relatively flat sea ice surfaces. Consequently, the filtering of these data is likely to bias high the freeboard height estimates. We therefore argue that these observations should




not be discarded and that the PP threshold used to identify sea ice floes should match the PP threshold used to identify sea ice leads. Hence, waveform echoes with a PP>0.3 are considered as lead observations while those with a PP<0.3 are considered as ice floe observations. In recent studies, the stack kurtosis has been employed in addition to the PP to distinguish leads from ice floes with CryoSat-2 (Ricker et al., 2014; Laxon et al., 2013). Unfortunately, the stack kurtosis cannot be estimated with

Envisat as LRM altimetry does not provide with multi-look angle observations. In the aim of keeping the freeboard height retrieval between Envisat and CryoSat-2 as similar as possible, we therefore do not make use of the stack kurtosis criterion and only use the PP to distinguish lead from ice floes observations.

To estimate the surface elevation of ocean and sea ice floes from the waveform echoes, several methodologies are found in the literature. In the ESA SI-CCI project, two different retracker algorithms are employed to take into account the difference of

waveform shape between specular leads and diffuse ice floes. In Laxon et al. (2013), only one retracker is used for ocean and ice floes observations but a constant correction is applied to take into account the difference of specularity between ocean and ice floes. In Ricker et al. (2014), the authors use a single retracker for ocean and ice floes but unlike Laxon et al. (2013), they do no use an extra-correction. Following the latter study, we only use a single retracker and make no extra-correction. This choice is made to limit the potential bias driven by the use of different algorithms and to fully understand the discrepancies between

Envisat and CryoSat-2 freeboard height measurements.

Over surfaces with an heterogeneous reflectivity such as sea ice, basic retracking algorithms based on the waveform maximum power can be easily biased by off-nadir reflexions. For this reason, we use the more robust Threshold First-Maximum Retracker Algorithm (TFMRA) to estimate the surface level position (Helm et al., 2014). For both Envisat and CryoSat-2 and for both leads and ice floes observations, the TFMRA retracker is parametrized identically.

The next step of freeboard height retrieval consists to interpolate the sea level underneath sea ice floes and to estimate the height of ice floes above the interpolated sea level. However, the sea level interpolation can drive large errors due to uncertainties in estimates of ocean tide, barometer tide and mean sea surface height. To minimize this error, we argue that freeboard height should be estimated in the surrounding of each lead within a section limited to 25 km after correcting the sea level from mean sea surface height, ocean tide and barometer tide. Single freeboard height measurements are then interpolated

along the altimeter track with a 25 km window median. Finally, monthly freeboard observations are smoothed onto a 12.5 km × 12.5 km polar stereographic grid using a median filter with a radius of 100 km. Similarly to the freeboard height maps, the Envisat along-track PP is also converted to monthly gridded maps.

### 2.4 Freeboard-to-thickness conversion

To convert the freeboard ($fb$) height to sea ice thickness (IT), we assume the hydrostatic equilibrium between snow-covered

sea ice and the ocean and we use the following expression:

$$IT = \frac{h_s\rho_s + fb\rho_i}{\rho_w - \rho_i} \tag{2}$$

where, $h_s$ represents snow depth, $\rho_i$ is ice density, $\rho_s$ is snow density and $\rho_w$ is sea water density.



To operate the freeboard-to-thickness conversion, several snow depth dataset have been employed. For instance, Laxon et al. (2003) and Giles et al. (2008) use the time and space-varying snow depth climatology (hereafter W99) from the study by Warren et al. (1999). However, recent *in situ* observations have demonstrated that snow depth has thinned by 50% over First Year Ice (FYI) relatively to the W99 climatology (Kurtz et al., 2013; Guerreiro et al., 2016). Following this result, Kwok and

Cunningham (2015) estimate snow depth according to the following equation:

$$h_s(X, t, f_{MY}) = h_s^{W99}(X, t)f_{MY} + 0.5h_s^{W99}(X, t)(1 - f_{MY}) \tag{3}$$

where $h_s(X, t, f_{MY})$ represents the time (t) and space (X)-varying W99 climatology and $f_{MY}$ is the Multi Year Ice (MYI) fraction that they derive from the Advanced Scatterometer (ASCAT) following the study by Kwok (2004). In the present study, we use a similar methodology to estimate snow depth except that we use a different dataset to derive MYI ratio. Indeed, the

methodology based on ASCAT observations assumes a direct link between ice age and ice roughness, which is not necessarily true in every case. To overcome this issue, we use the National Snow and Ice Data Center (NSIDC) ice age dataset (http://nsidc.org/data/docs/daac/nsidc0611-sea-ice-age/, Anderson and Tschudi (2014)). In this product version, the ice age is calculated by tracking the ice through the comparison of adjacent passive microwave images as well as using wind forcing and buoys displacement information. In addition to not make the assumption of a constant correlation between ice age and ice

roughness, this product has the advantage to be available during the entire Envisat mission (2002-2012). In the present study, monthly snow depth fields are therefore derived as follows: First, each ice pixel with ice older than 1 year is considered as MYI and pixels with ice younger than 1 year is considered as FYI. Then, we apply an average filter with a radius of 100 km (similarly to the freeboard datasets) and estimate the ratio of MYI observation within the filtering area to build MYI concentration maps. Finally, we use the resulting MYI ratio fields, the W99 climatology and Eq. 3 to determine monthly snow depth fields.

An example of snow depth field is provided in Fig. 1 for March 2011.

Based on the results by Alexandrov et al. (2010), recent studies attribute a lower density to MYI (882 $kg.m^{-3}$) than to FYI (917 $kg.m^{-3}$). In their study, Timco and Frederking (1996) show that the density of MYI is indeed lower than the density of FYI for samples obtained above the waterline but that the density of FYI and MYI are not significantly different for samples obtained bellow the waterline. Based on this result and on the fact that most of the ice ($\sim 90\%$) is found bellow the waterline,

we argue that a single value should be used to parametrize sea ice density. Hence, after testing several ice density values (see results in section 3.4), we use a single ice density regardless the ice age.

Regarding snow and sea water densities, we use the same parametrization as in Kwok and Cunningham (2015). More precisely, the snow density follows the monthly average prescribed by Warren et al. (1999) and the sea water density is set to 1024 $kg.m^{-3}$.

An another important step for the freeboard-to-thickness conversion is the correction of the slower wave propagation effect in the snow pack. Indeed, the radar signal propagates slower in snow than in the atmosphere, which drives an underestimation of





the freeboard height estimates. To correct this bias, we add the factor $\alpha$ (see Eq. 4) to correct the freeboard height measurements as operated in Kwok and Cunningham (2015).

$$\alpha = h_s(1 + 0.5\rho_s)^{-1.5} \tag{4}$$

## 2.5 Ancillary data

Daily sea ice extent fields are available at a 12.5 km resolution from NSIDC and are derived from AMSR-E and SSMIS passive microwave observations (Anderson and Tschudi, 2014). The ice extent dataset is used to grid the freeboard height observations in regions where sea ice is present and to remove observations over open water areas.

The Beaufort Gyre Exploration Project (BGEP) moorings provide high frequency (0.5 Hz) measurements of sea ice draft since August 2003 thanks to a network of 2 to 4 moorings located in the Beaufort Sea (http://www.whoi.edu). The buoys are equipped among other instruments of an Upward Looking Sonar (ULS) that measures the distance from the mooring to the bottom of the ice with a 420 kHz beam sonar. Sea ice draft is then estimated by removing the depth of the mooring that is deduced from pressure measured every 40 seconds. The accuracy of each 0.5 Hz measurements is of ± 5 cm. The position of each of the four moorings used in the present study is shown with a black square in Fig. 1.

## 3 Results

### 3.1 Comparison of CryoSat-2 and Envisat freeboard height

In Fig. 2, we show monthly freebard height maps from November 2010 to April 2011 estimated with CryoSat-2 (Fig. 2a) and Envisat (Fig. 2b). The parameter shown in Fig. 2c will be introduced in section 3.3. Note that due to the spatial smoothing, freeboard observations are available even in SARIn regions. First of all, most Envisat freeboard height observations are negative (-13 cm in average), which implies that the sea level is observed above the level of sea ice floes. These unrealistic observations suggest that the difference of ice surface characteristics between leads and ice floes as well as the use of a threshold retracker drive a large bias on the estimation of Envisat freeboard height. Unlike Envisat, CryoSat-2 displays an average freeboard height (3.5 cm) consistent with previous studies (Armitage and Ridout, 2015; Kurtz et al., 2014), which demonstrates that despite the ice surface diffusion difference between leads and ice floes, CryoSat-2 allows to produce a realistic freeboard height dataset.

Regarding the seasonal variability, Envisat displays an unrealistic negative variation of -2.4 cm cm from November 2010 to April 2011 and -3.7 cm from November 2011 and March 2012 (see Tab. 1). Only ice located north of the Canadian Archipelago displays a significant variation from November 2010 (-20 cm) to April 2011 (-7 cm). Unlike for Envisat, CryoSat-2 freeboard height displays a significant seasonal cycle with an average variation of 2.7 cm from November 2010 to April 2011 and 2.4 cm from November 2011 to March 2012 and a growth visible in all Arctic regions.



## 3.2 Impact of ice surface diffusion on the CryoSat-2 and Envisat freeboard height discrepancy

We now seek to analyze more in details the discrepancy of freeboard height between Envisat and CryoSat-2 and its linkage with the ice surface diffusion. In Fig. 3a, we show monthly maps of the freeboard height difference ($\Delta$fb) for the 2010/2011 ice growth season between Envisat and CryoSat-2 (ENVISAT - CryoSat-2). For every Arctic region and for each month of the

period of study, CryoSat-2 is always thicker than Envisat ($\Delta$fb <0). This result differs from the study by Schwegmann et al. (2015) due to the fact that the authors use several waveform retrackers to retrieve Envisat freeboard height while we only use a single retracker. During the entire ice growth season, high $\Delta$fb observations are located over FYI in marginal regions as in the Bering Strait, the Baffin Bay, at north of the Atlantic Ocean and in coastal regions. Furthermore, high $\Delta$fb are particularly present at the beginning of the ice growth season. Conversely, low $\Delta$fb observations are mainly located over MYI areas in

the North Canadian Archipelago and are mostly visible at the end of winter. The evolution of the $\Delta$fb all along the winter season suggests that the discrepancy between CryoSat-2 and Envisat freeboard height is related with changes of ice surface characteristics.

    To analyze the link between ice surface diffusion and the freeboard height discrepancy between Envisat and CryoSat-2, the shape of the radar waveform echoes can be used. Indeed, surface diffusion due to ice roughness and snow volume scattering

modifies the shape of waveform echoes, which allows to identify ice surface characteristics. Examples of typical Envisat waveforms are provided in Fig. 4. For specular surfaces, the radar signal back-scattered toward the satellite originates from a small surface around the nadir position (Connor et al., 2009) and most of the back-scattered signal reaches the radar altimeter within a relatively short delay yielding to a specular waveform echoes (Fig. 4d). For diffuse surfaces, the proportion of off-Nadir signal increases and the radar altimeter receives the back-scattered signal during a longer period, which yields to a more

diffuse waveform echoes (Fig. 4a, b and c). To characterize the waveform echoes diffusion and thus the ice surface diffusion, the PP can be used (Zygmuntowska et al., 2013). Indeed, high values of PP characterize specular waveform echoes while lower values of PP characterize diffuse echos as shown in Fig. 4.

    In Fig. 3b, we show monthly maps of Envisat PP for the 2010/2011 ice growth season. In November, most Arctic sea ice has a PP larger than 0.1. These relatively high values are mainly explained by the presence of specular young sea ice and/or melted

snow that act as a mirror for nadir looking altimeters. Only rough MYI areas display PP observations lower than 0.1 during this period of the year. The average Envisat PP decreases progressively from 0.14 in November to 0.06 in April due to the ice growth, ice deformation and snow thickening that increase the ice surface diffusion. Looking at Fig. 3a and 3b, it is striking how the PP and the $\Delta$fb maps are correlated. This relation between the two parameters is particularly visible in marginal ice regions and in near-coastal regions where the PP and the $\Delta$fb are both relatively high. To further demonstrate the link between

the PP and the $\Delta$fb, we show in Fig. 5 the relation between the $\Delta$fb and the Envisat PP for November 2010-April 2011 (blue) and November 2011-March 2012 (red) as well as the polynomial regression obtained with the entire set of observation (dark dashed line). As suggested by the visual observations of Fig. 3, there is a clear correlation between the PP and the $\Delta$fb (R = 0.91). The freeboard height difference varies from -35 cm for low PP to -5 cm for high PP. Between the two ice growth seasons, the relation between the PP and the $\Delta$fb is fairly identical despite a slight discrepancy for low PP values (<0.05) and





high PP values (>0.25). These discrepancies are mainly driven by the lower amount of marginal PP observations as well as by the relatively high deviation of Δfb observations caused by the asynchronous freeboard height measurements.

Despite this slight difference between the two winter periods, the strong linkage between the PP and the Δfb shown in Fig. 3 and Fig. 5 supports the hypothesis that the discrepancy between the Envisat and CryoSat-2 freeboard height datasets is
related to the variability of ice surface diffusion. Moreover, the unrealistic Envisat freeboard height estimates in contrast to the accurate CryoSat-2 estimates suggests that the discrepancy between Envisat and CryoSat-2 freeboard height fields is mainly driven by a larger impact of ice surface diffusion on Envisat waveform echoes as postulated in the introduction section. This results demonstrates therefore that it is essential to take into account the impact of ice surface diffusion on Envisat waveform echoes to improve the freeboard height estimates.

**3.3 Toward an improved Envisat freeboard: Envisat/PP**

Over oceanic surfaces, the Brown model (Brown, 1977) is generally used to take into account the effect of ocean surface diffusion and to improve the estimation of sea level. Over sea ice, the diffusion variability is too high and the Brown model cannot be applied. Consequently, threshold retrackers are used instead in most sea ice studies. However, it has been shown recently that the use of threshold retrackers over diffuse ice can drive biases on the freeboard height retrieval (Kurtz et al.,
2014). Due to the reduced footprint of SAR altimetry, the bias caused by the use of threshold retrackers is not so critical for CryoSat-2 as shown by the comparison of ice thickness estimates with *in situ* observations (Laxon et al., 2013; Kwok and Cunningham, 2015). However, ice surface diffusion has a higher impact on LRM altimeters as suggested in the present study, and the current Envisat freeboard estimates are strongly biased. To improve the Envisat freeboard height retrievals, a first approach could consist in developing a retracking algorithm model that takes into account the large variability of ice surface
diffusion as operated in Kurtz et al. (2014), instead of using threshold retrackers. While such model could provide accurate measurement of sea ice freeboard height, it requires accurate knowledge on sea ice characteristics such as mean surface slope (diffusion), angular backscattering efficiency (specularity) and snow volume scattering properties, which are currently hardly measurable parameters. Another approach to improve the Envisat freeboard height retrievals would consist in correcting the Envisat freeboard dataset with the results obtained in the present study. In particular, the regression curve shown in Fig. 5 can
be used to build a CryoSat-2-like version of Envisat freeboard height. We acknowledge that the "PP-correction" would not entirely remove the bias related to the ice surface diffusion variability as CryoSat-2 freeboard height is also likely impacted by the variability of ice surface diffusion (Kurtz et al., 2014). Nevertheless, such approach would allow to obtain freeboard height estimates during the entire Envisat mission with an accuracy similar to CryoSat-2 estimates (Laxon et al., 2013; Kwok and Cunningham, 2015), which could strongly benefit to the study of long term ice thickness variations. Following the latter
approach, we show in Fig. 2c the Envisat freeboard height corrected as follows:

$$fb_{pp} = fb - y(PP) \tag{5}$$



Where $fb_{pp}$ is the corrected Envisat freeboard height (hereafter Envisat/PP), PP is the Envisat pulse-peakiness, $fb$ is the un-corrected Envisat freeboard height and and $y(PP)$ is the average relationship between PP and $\Delta fb$ shown in black in Fig. 5. First of all, the Envisat/PP dataset displays a realistic average freeboard height during the entire winter (see Tab. 1). Looking at Fig. 2a and 2c, Envisat freeboard height is very similar to the results obtained with CryoSat-2, with identical patterns

in most Arctic regions. Only ice located north of the Candian Archipelago displays a discrepancy with a thicker CryoSat-2 freeboard height. This discrepancy is due to the higher uncertainty on CryoSat-2 and Envisat freeboard height estimates over very diffuse surfaces as shown in Fig. 5 for low PP observations. In Tab. 2, we show the freeboard height RMSD between Envisat and CryoSat-2 and between Envisat/PP and CryoSat-2. The Envisat/PP dataset displays a clear improvement with an average RMSD of ∼2 cm in contrast to the uncorrected Envisat dataset (∼16 cm). In addition to the spatial variability, the

seasonal variation is clearly improved with a variation of 3.0 cm between November 2010 and April 2011 and of 2.0 cm between November 2011 and March 2012. These results demonstrate therefore that the relation between the Envisat PP and $\Delta fb$ allows to build a robust CryoSat-2-like version of Envisat freeboard height during each month of the common flight period of Envisat and CryoSat-2 missions.

### 3.4   Comparison to BGEP ice draft measurements

To assess the potential of the PP-correction approach to produce accurate ice thickness estimates, we now convert the CryoSat-2, Envisat and Envisat/PP freeboard height datasets to sea ice draft (thickness - freeboard) fields and compare the results to BGEP ice draft measurements. For this purpose, we estimate the monthly median altimetric ice draft within a radius of 50 km around each available mooring, which we compare to the corresponding monthly median mooring ice draft. To obtain accurate ice draft estimates, we first test three different sea ice densities (890, 900 and 910 $kg.m^{-3}$) to convert CryoSat-2 freeboard

height (see Tab. 3). For the three density values, the correlation coefficient is similar (0.83) but the average RMSE is fairly different depending on which density value is used. As suggested by Tab. 3, the best results are obtained with an ice density of 900 $kg.m^{-3}$ (RMSE = 0.2 m). Following this result, we use the 900 $kg.m^{-3}$ value for all type of ice and for both CryoSat-2 and Envisat datasets to convert freeboard height to ice draft.

In Fig. 6, we show the monthly CryoSat-2 and Envisat/PP ice draft estimates as a function of the corresponding monthly

BGEP ice draft. In color, we represent the MYI fraction for each observation. We also show in gray the uncorrected Envisat ice draft estimates. The CryoSat-2 ice draft estimates have a relatively low RMSE (0.13 m) with *in situ* measurements during the 2010/2011 ice growth season (Fig. 6a) and a higher RMSE (0.25 m) during the 2011/2012 ice growth season (Fig. 6b). The higher RMSE observed in Fig. 6b is mainly driven by a few underestimated values characterized by a very low MYI fraction. This result suggests that the higher RMSE obtained during 2011/2012 might be caused by an error in the snow depth

parametrization rather than by an error in the freeboard height fields, which highlights the need of improving snow depth fields to improve the freeboard-to-thickness conversion.

The comparison of the Envisat/PP ice draft with the uncorrected Envisat PP during the 2010/2012 period clearly shows the improvement brought by the PP-correction. The correlation between the Envisat/PP dataset and the BGEP ice draft measurements is similar as for CryoSat-2 with a low RMSE (0.16 m) during winter 2010/2011 (Fig. 6j) and a higher RMSE (0.29 m)





during winter 2011/2012 (Fig. 6k). This result demonstrates therefore the potential of the PP-correction to produce Envisat ice draft estimates as accurate as CryoSat-2 during the entire cross-calibration period.

To further assess the potential of the PP correction, it is necessary to verify if the Envisat/PP dataset is also valid out of the cross-calibration period. As the BGEP ice draft measurements are available from August 2003, the accuracy of the PP-

correction can be evaluated over most of the Envisat mission lifetime. The Envisat and Envisat/PP ice draft estimates are therefore computed over the 2003-2010 ice growth seasons and are compared with the corresponding mooring observations (Fig. 6c-i). As during the 2010-2012 period, the Envisat/PP ice draft dataset displays a good agreement with the buoys obser-vations (RMSE =0.16 to 0.30 m) relatively to the uncorrected Envisat dataset (RMSE = 1.06 to 1.43 m). In Fig. 6c,h and i, underestimated values are characterized by a relatively low MY fraction. As for the 2010/2012 period, this result suggests that

the error on ice draft estimates is likely driven by a inaccurate construction of the snow depth fields. Despite these punctual discrepancies, the good agreement between the Envisat/PP and the BGEP ice draft estimates demonstrates that the Envisat/PP dataset provides accurate estimates of sea ice draft during the entire Envisat mission.

## 3.5 Time series of circumpolar ice thickness and volume

As shown in the previous sections, the PP-correction allows to obtain an improved Envisat freeboard height dataset over

the entire Envisat mission. Furthermore, the PP-correction allows to cross-calibrate Envisat and CryoSat-2 freeboard height estimates and enable to combine the two datasets to produce a time series of ice thickness over more than a decade. In Fig. 7, we show the average circumpolar sea ice thickness from 2002 to 2015 estimated with Envisat/PP (blue) and CryoSat-2 (red). It is the first time that pan-Arctic sea ice thickness is estimated during more than a decade. It is also the first time that pan-Arctic ice thickness is presented during the 2008/2009 and 2009/2010 ice growth seasons as neither CryoSat-2 nor ICESat (Zwally

et al., 2002) missions provide results during this period. As expected, the seasonal ice thickness variation between CryoSat-2 and Envisat during the 2010-2012 period is fairly identical for the 2 satellite missions, despite a higher estimation shown by Envisat at the beginning of the 2011/2012 winter.

As shown in Kwok et al. (2009) and Giles et al. (2008), the average pan-Arctic and winter ice thickness is relatively thick (> 1 m) during the 2002-2007 period, with a maximum in 2002/2003 (1.11 m), and then strongly decreases during winter

2007/2008 (0.84 m). This large thinning was attributed to the strong summer melt that occurred in 2007 and to the resulting loss of thick MYI (Lindsay et al., 2009). From 2008 to 2010, the average circumpolar ice thickness increases back but remains thinner than 1 m. From 2010 to 2012 the circumpolar sea ice thickness decreases again and reaches its lowest level (0.83 m). As in 2008, this strong thinning is most likely caused by the strong summer melt that occurred in 2012 and to an important loss of thick MYI (Parkinson and Comiso, 2013). Over the most recent period (2012-2015) the circumpolar ice thickness

strongly increases and recovers a level similar as during the 2002-2007 period with ice thicker than 1 m in average. This strong thickening of Arctic sea ice was attributed to the retention of thick sea ice north of the Canadian Archipelago during 2013 as well as to a 5% drop in the number of days on which melting occurred (Tilling et al., 2015).





## 4  Discussion

In the present study, we investigate the inconsistency between Envisat and CryoSat-2 freeboard height. We show that ice surface diffusion drives most of the difference between the two freeboard height datasets and that a correction based on the waveform PP can be used to improve Envisat freeboard height estimates. While the PP-correction allows to remove most of

the bias between CryoSat-2 and Envisat freeboard height datasets, other sources of inconsistencies could explain the remaining difference between the 2 datasets. For instance, bright off-nadir reflexions have a higher impact on Envisat than on CryoSat-2. The reduced footprint of CryoSat-2 allows to filter out bright off-nadir reflections in the along-track direction, which enables to produce more accurate estimates of surface level position than with Envisat. As this discrepancy between the two radar altimeters is not corrected by the PP-correction, the Envisat/PP freeboard height is likely less accurate than CryoSat-2's.

Nevertheless, the relatively low RMSD observed between Envisat/PP and CryoSat-2 freeboard height (see Tab. 2) suggests that the bias driven by off-nadir reflections on the Envisat/PP freeboard height should be negligible.

Another source of inaccuracy of the Envisat/PP dataset is the function used to correct the Envisat freeboard height (y(PP) in Eq. (5)). This function is obtained by fitting the freeboard height difference between CryoSat-2 and Envisat with the Envisat PP and is used to produce a CryoSat-2-like version of Envisat freeboard height. Thus, the accuracy of the Envisat/PP freeboard

height depends strongly on the accuracy of the CryoSat-2 dataset. In the present study, CryoSat-2 freeboard height is processed with a threshold retracker. However, in the study by Kurtz et al. (2014), the authors argue that the use of a threshold retracker could drive biases in CryoSat-2 freeboard height estimates due to the impact of ice surface diffusion. Having said that, the comparison of our CryoSat-2 ice draft estimates with *in situ* measurements as well as preliminary studies (Laxon et al., 2013; Kwok and Cunningham, 2015) demonstrate that CryoSat-2 freeboard height estimates are fairly accurate despite the use of a

threshold retracker. Consequently, the impact of CryoSat-2 freeboard height uncertainty on the Envisat/PP estimates should be minor.

## 5  Conclusions

In this study we investigate the inconsistency between CryoSat-2 and Envisat freeboard height during the common flight period (2010/2012). During the entire period of study, Envisat freeboard height is always thinner (-13 cm) than CryoSat-2 freeboard

height (3.5 cm). Furthermore Envisat freeboard height displays an unrealistic negative seasonal variation (-2.4 to -3.7 cm) while CryoSat-2 freeboard height displays a realistic positive variation (2.4 to 2.7 cm). The comparison between the freeboard height discrepancy and the Envisat PP demonstrates that these discrepancies are linked with the impact of ice surface diffusion driven by ice roughness and snow volume scattering. The freeboard height difference is found to vary from -30 cm to -5 cm for a PP varying between 0.05 and 0.3. The agreement between CryoSat-2 and *in situ* ice draft estimates in contrast to the unrealistic

Envisat freeboard height estimates suggests that the discrepancy between the two datasets is most entirely driven by a higher sensitivity of Envisat freeboard height to ice surface diffusion variability.

To correct the bias due to ice surface diffusion variability on the Envisat freeboard height estimates, we build an improved Envisat freeboard height dataset (Envisat/PP) based on the relation between the freeboard height difference between Envisat





and CryoSat-2 and the Envisat PP. The resulting Envisat/PP freeboard estimates displays similar patterns than CryoSat-2 during the entire period of study (RMSD = ∼ 2 cm) and offers a more realistic seasonal cycle (∼ 2.0 to 3.0 cm) than the uncorrected Envisat freeboard.

The comparison of each altimetric dataset with *in situ* ice draft observations during the common flight period reveals that Envisat/PP ice draft is highly more accurate (RMSE = 0.16 - 0.29 m) than the uncorrected Envisat dataset (RMSE = 0.61 - 0.69 m) and as accurate as CryoSat-2 (RMSE = 0.13 - 0.25 m). To further assess the potential of the PP-correction, the Envisat and Envisat/PP ice draft datasets are extended to the 2003-2010 period and are similarly compared to BGEP ice draft measurements. As for the 2010-2012 period, the agreement with *in situ* measurements is higher for Envisat/PP (RMSE = 0.14 - 0.29 m) than for Envisat (RMSE = 1.08 - 1.32 m), which demonstrates the potential of the PP-correction to provide accurate freeboard height estimates over the entire Envisat mission. Another important result of this comparison is the strong ice thickness uncertainties related to inaccurate snow depth parametrization.

A rapid analysis of the circumpolar Arctic ice thickness during the 2002-2015 period reveals that Arctic sea ice has considerably thinned in 2007/2008 and 2012/2013 following strong summer melt periods. Over the most recent period (2013-2015), the average pan-Arctic ice thickness has recovered a level similar a during the 2002-2007 period.



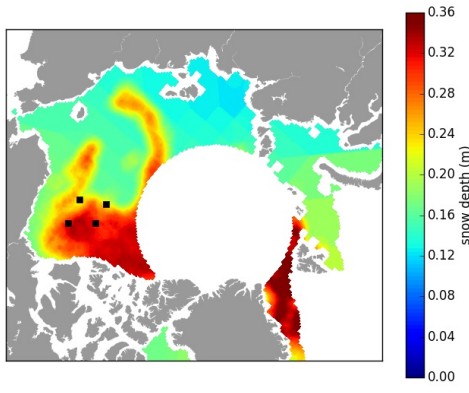

**Figure 1.** Monthly snow depth map shown for March 2011 and obtained from the NSIDC ice age dataset. The dark squares indicate the position of the BGEP moorings.





**Figure 2.** a) CryoSat-2 , b) Envisat and c) Envisat/PP monthly freeboard height maps shown for the November 2010 - April 2011 period. The dark lines represent the isoline fraction of MYI equal to 0.7.

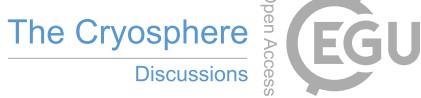



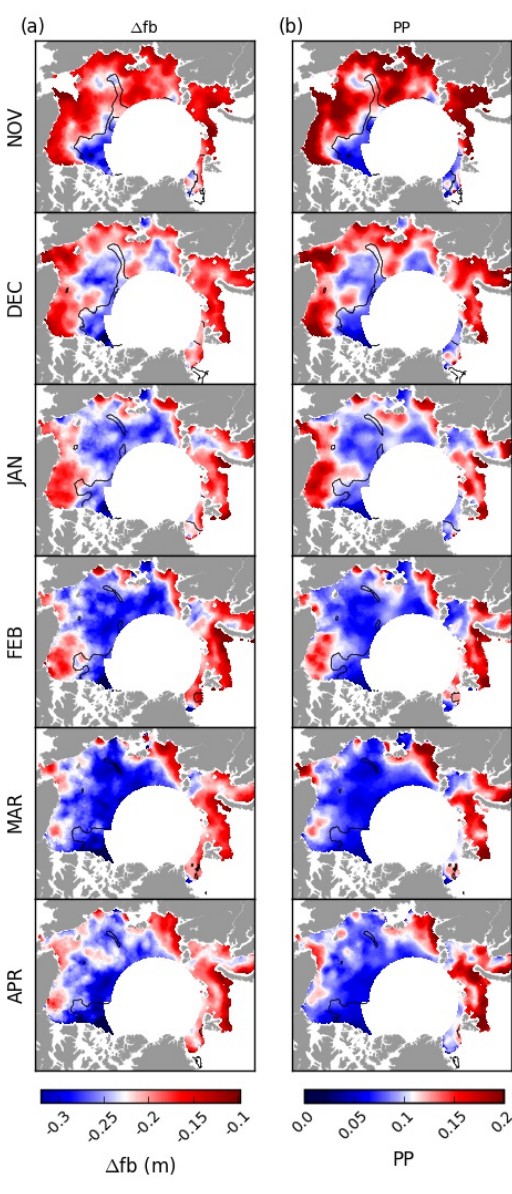

**Figure 3.** a) Monthly freeboard height difference (Δfb) between Envisat and CryoSat-2 and b) monthly Envisat pulse peakiness (PP) during the November 2010 - March 2012 period. The dark lines represent the isoline MYI fraction equal to 0.7





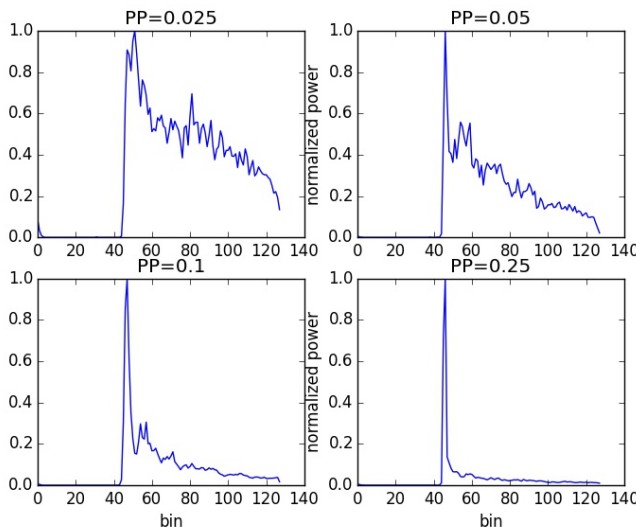

**Figure 4.** Typical Envisat normalized waveform echoes over sea ice for various values of PP (0.05, 0.1, 0.2, 0.4)

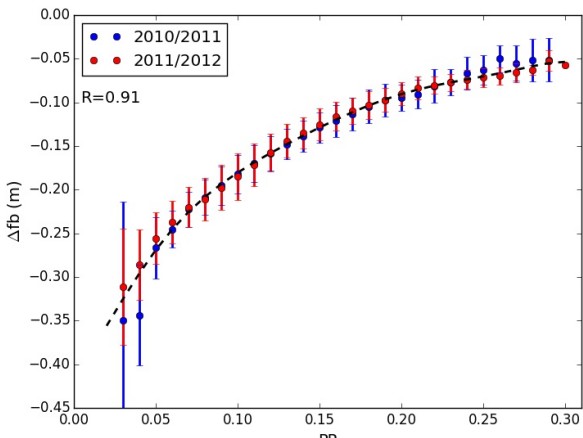

**Figure 5.** Relation between the monthly freeboard height difference between Envisat and CryoSat-2 (Δfb) and the monthly Envisat pulse-peakiness (PP) shown for winter 2010/2011 (blue) and 2011/2012 (red). The error bars show the corresponding standard deviation for each value and the dark tilted line represents the fitted curve of all monthly observations during the entire period of study (November 2010 - April 2011 and November 2011 - March 2012).




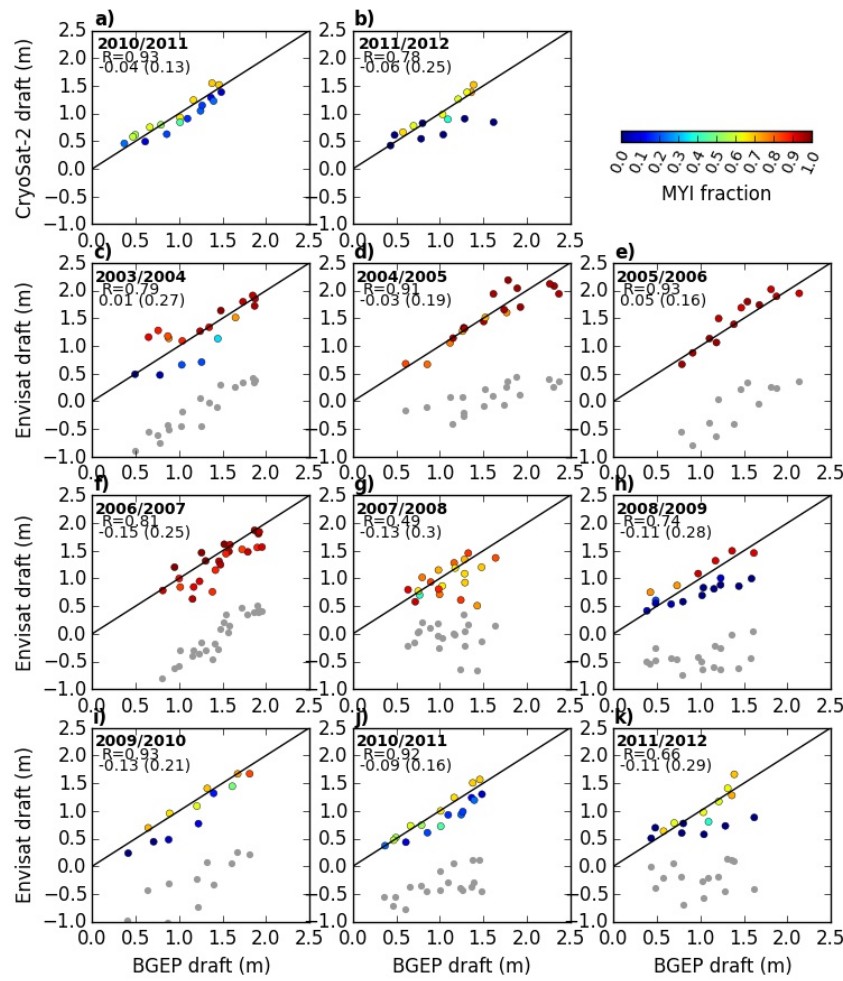

**Figure 6.** CryoSat-2 monthly ice draft as a function of BGEP moorings monthly median ice draft for winters (November to April) 2010/2011 (a) and 2011/2012 (b). Envisat (gray) and Envisat/PP (colored) monthly ice draft as a function of BGEP moorings monthly median ice draft for winters 2003/2004 to 2011/2012 (c-k). The color scale used for CryoSat-2 and Envisat/PP datasets shows the corresponding Multi Year Ice (MYI) fraction. For each winter, the correlation coefficient (R), the average bias and the RMSE (in parenthesis) is shown. The number of observations for each year depends on the corresponding number of available BGEP buoys (from 2 to 4).



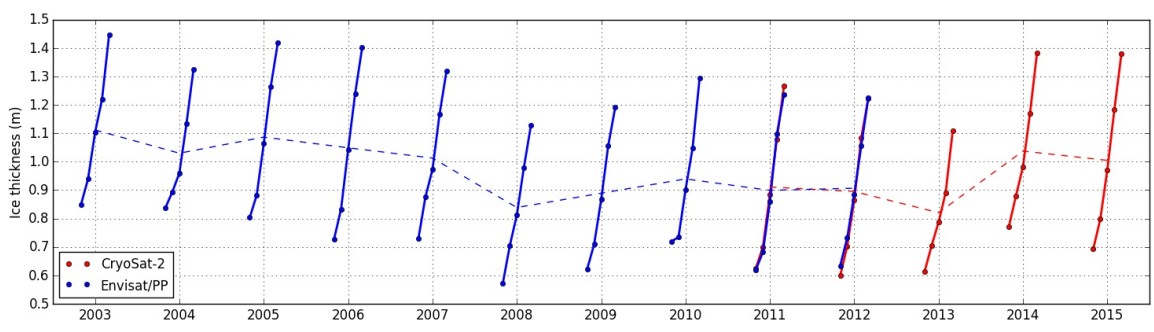

**Figure 7.** Mean monthly pan-Arctic sea ice thickness from Envisat/PP (blue) and CryoSat-2 (red) given from November 2002 to March 2015. The titled lines shows the Pan-Arctic winter average (November-March) sea ice thickness.





**Table 1.** Monthly average pan-Arctic freeboard height (in centimetres) for November 2010 - April 2011 and November 2011 - March 2012 given for Envisat, CryoSat-2 and Envisat/PP

|  | Envisat | CryoSat-2 | Envisat/PP |
|---|---|---|---|
| 2010/2011 |  |  |  |
| NOV | -11.4 | 2.2 | 2.2 |
| DEC | -13.5 | 2.5 | 2.3 |
| JAN | -14.3 | 3.2 | 2.9 |
| FEB | -14.1 | 4.2 | 4.4 |
| MAR | -14.5 | 5.1 | 4.8 |
| APR | -13.5 | 4.9 | 5.2 |
| 2011/2012 |  |  |  |
| NOV | -10.2 | 2.0 | 2.4 |
| DEC | -11.8 | 2.3 | 2.7 |
| JAN | -12.9 | 2.7 | 2.9 |
| FEB | -14.4 | 4.0 | 3.6 |
| MAR | -13.9 | 4.4 | 4.4 |

**Table 2.** Monthly root mean square difference (centimetres) between CryoSat-2 and Envisat (1st column) and between CryoSat-2 and Envisat/PP (2nd column) for November 2010 - April 2011 and November 2011 - March 2012.

|  | Envisat | Envisat/PP |
|---|---|---|
| 2010/2011 |  |  |
| Nov | 14.0 | 1.5 |
| Dec | 16.4 | 1.6 |
| Jan | 18.0 | 2.0 |
| Feb | 19.1 | 2.5 |
| Mar | 20.3 | 2.4 |
| Apr | 19.1 | 2.2 |
| 2011/2012 |  |  |
| Nov | 13.2 | 1.4 |
| Dec | 14.9 | 1.6 |
| Jan | 16.3 | 2.1 |
| Feb | 19.1 | 2.8 |
| Mar | 19.2 | 2.7 |

*Acknowledgements.* This research is supported by the French CNES TOSCA SICKAyS. We thank the Center for Topographic studies of the Oceans and Hydrosphere (CTOH) at LEGOS for providing the SGDR Envisat and CryoSat-2 data.



**Table 3.** Average monthly ice draft correlation coefficient (R) and Root Mean Square Error (RMSE) of CryoSat-2 ice draft during winters 2010/2011 and 2011/2012 tested for 3 different ice density values (890, 900 and 910 $kg.m^{-3}$).

|  | **890** $kg.m^{-3}$ | **900** $kg.m^{-3}$ | **910** $kg.m^{-3}$ |
|---|---|---|---|
| R | 0.83 | 0.83 | 0.83 |
| RMSE | 0.22 cm | 0.20 cm | 0.23 cm |



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
