# Peer review of "Comparison of CryoSat-2 and ENVISAT freeboard height over Arctic sea ice: Toward an improved Envisat freeboard height retrieval."

_The Cryosphere, 2016_

## Referee Comment (RC1) · EJ Rinne (Referee) · 15 Feb 2017

The manuscript presents a time series of circumarctic sea ice freeboard (and thickness) from two satellite altimeters. This is an extremely relevant for TC and significant and the paper is reasonably well written, although lacks detail in some important parts. The authors address the problem of combining measurements of two different altimeters (Cryosat-2 being delay-doppler and Envisat RA-2 not) by introducing a correction to Envisat freeboards based on Pulse Peakiness, which is a novel and original idea that has never been published before. However, there is a striking problem with the Envisat freeboard estimates before the PP correction is applied, and thus I urge the authors to fix their Envisat base methodology before this paper is considered for publication.

Furthermore, even if the PP correction is shown to produce very good agreement with Cryosat-2 and BGEP data, the theoretical justification of the correction is lacking. Both of these two issues should be addressed before the manuscript is published in TC and my suggestion is that the manuscript should go through major revision before accepted.

The major issues in the manuscript are:

1. The Envisat FB product before PP correction

Looking at figure 2 middle column, one can see that the Envisat freeboards are unrealistic. For one, they are negative – something that the authors just attribute to "the difference of ice surface characteristics between leads and ice floes as well as the use of a threshold retracker drive a large bias on the estimation of Envisat freeboard height". I am confident that the culprit is elsewhere.

We've tested the TFMRA retracking scheme for Envisat as well in the CCI project, and we've arrived at more or less similar looking freeboard maps as with the original CCI retracking scheme. We seem to be missing the thinnest and the thickest ice, but freeboards are positive as they are supposed to be and the thickness pattern reflects reality (even with the thinnest and thickest ice missing). And furthermore, we do not see very high freeboards in the marginal ice zone.

I try to be a good reviewer and speculate possible causes for the Envisat freeboards being much off. My guess is that this may be due to off-nadir leads or new ice dominating significant number of waveforms. The authors give very little notice to filtering out mixed waveforms. Or filtering in general – it is hardly mentioned anywhere in the paper. They argue that they should keep in the waveforms with intermediate PP since they represent thin and undeformed ice. Fair enough, but at the same time they are letting in a lot of waveforms with deformed ice in the nadir and flat areas off-nadir which will lead into the retracker catching the off-nadir rise and biasing the elevation estimate low. This is consistent with the lowest freeboards seen in the area with lot of deformed ice (there will always be a significant number of flat new ice or leads around). This is

less of a problem in the area of new ice near the margins, where the ice is more or less flat all around and in likelihood there is a specular surface in the nadir as well. All consistent with the pattern in figure 2.

The authors hint that the use of TFMRA retracker is robust for off nadir reflections (page 5, lines 16-19). That is somewhat true, but it does not remove the need to filter out dubious waveforms – even Helm et al 2014 that the authors cite for the TFMRA have a filtering scheme to remove "bad waveforms" before retracking. I suggest the authors build one too and check if that improves their not-PP corrected freeboards. The SI-CCI scheme most likely filters too much waveforms, but I would still argue that some kind of filtering is required.

Finally , much less likely culprit than previous one , but worth mentioning still since applying an inverted snow correction (that is, a bug in code) results into something bit like the maps in Figure 2. The main reason I'm mentioning this is that I once had that bug in my code and the Figure 2 reminds me much of it. Don't waste too much time on this, but do check your snow propagation correction code.

2. Theoretical justification of the PP correction

The manuscript fails to explain the theoretical background of why exactly small PP (or more diffuse waveforms or heavily deformed ice) results into retracker picking up the tracking point later in the waveform that it would if the waveform was more peaky (less diffuse and most likely originating from less deformed ice). The authors state that "ice surface diffusion has a higher impact on LRM altimeters" but this needs to be backed up by something solid because from the evidence authors give. Because of the unrealistic Envisat FB, I do not believe that the disagreement of the Envisat and CS-2 freeboards is mainly due to surface diffusion. If the authors do not, theoretically step by step, explain the process of ice surface diffusion impacting LRM altimeter estimates, a good referee could (and should) claim that it is just as likely that what we are seeing here instead is something profoundly wrong with the Envisat FB retrieval and that something

is connected to pulse peakiness.

The y(PP) is problematic anyway. Naturally, applying any correction derived from the difference of the two freeboard datasets will make the two agree. Strongest point the authors give for the use of the y(PP) correction is the improvement it brings to the fit of BGEP data throughout the Envisat period. This is all good and well, but looking at figure 6, the only real improvement is the level correction of about 1 – 1,5 m to the (unrealistic and often even negative) Envisat draft estimates. I would argue that what we see here is the constant term of y(PP) – there must be one since the dashed line in figure 5 does not cross zero – just fixes the large negative bias that the somehow broken Envisat freeboard method produces.

Furthermore one could argue that there is a relationship between PP and ice thickness. Thicker the ice, more deformation there is, thus rougher the surface and finally smaller PP. I find it likely that one would get reasonable results if they would just set thickness to be a function z(PP) only by fitting arctic-wide freeboard maps (from CS-2) to Envisat PP. One would also get a nice seasonal pattern of thickness since PP will go down through the winter with thicker ice and more deformation. Thus we do not even need retracking process to explain why introducing a PP correction improves the fit to independent data.

After the harsh critique above, I should mention that the idea presented in the manuscript is most definitely on the right track! A PP based correction would improve the problems of Envisat FB retrieval drastically. I know of similar attempts in the altimetric community lately. After fixing their uncorrected FB estimates and giving a theoretical justification of how the correction works, this will be a really good paper and I commend the authors for coming up with the idea and publishing it first. Problem with the manuscript at the moment is, that even if the final result of corrected Envisat freeboards seems to comply with validation data, the paper fails to give rigorous explanation exactly what are the processes why their methodology works.

Minor issues:

1. Pan-arctic claim

The authors claim that they have created a pan-Arctic thickness estimate. They have not, since they have excluded all of the Arctic above 81,5 N. Thus I recommend the authors follow the lead of Giles et al and stick with "circumpolar" (or something similar) to emphasise that their estimate does not cover all of the Arctic.

2. TFMRA parameters

Nowhere in the paper the authors state, which threshold value they use for the TFMRA. 50%? It should be mentioned. Like other TFMRA parameters as well.

3. Local sea level interpolation

The description of the sea level interpolation is thin (page 5, lines 20 – 24). Not sure if the interpolation of leads could contribute to the unrealistic negative freeboards, but it is worth checking. Nevertheless, the authors must include a better description of the sea level interpolation – how exactly is it done? Taking a mean of all lead elevations within 25 km or some kind of along-track interpolation?

4. PP correction – are leads included?

On page 5, line 27 it is stated that PP is also averaged into gridded maps. Does this include the waveforms that are classified as leads? If it does, this will have a consequence to the PP correction – that is, areas with lot of leads will eventually have a stronger correction in the direction of thinner ice.

5. Mathematical description of y(PP)

The authors really must give a more thorough description of the y(PP). Is the y(PP) constant throughout the winter? I reckon it is the black dashed line in Figure 5, and constant over time and place and calculated on the gridded level and not for individual measurements, but a mathematical formulation would be most welcome.

6. A typo:

Page 10 line 5: Candian

---

## Referee Comment (RC2) · Anonymous Referee #2 · 15 Feb 2017

Review for "Comparison of CryoSat-2 (CS2) and ENVISAT (ES) freeboard height over Arctic sea ice: Toward an improved Envisat freeboard height retrieval" by Guerreiro et al.

General comments:

The study "Comparison of CryoSat-2 (CS2) and ENVISAT (ES) freeboard height over Arctic sea ice: Toward an improved Envisat freeboard height retrieval" compares CryoSat-2 and Envisat freeboard retrievals, certainly for the overlap period 2010-2012. The authors use geo-located CS2 and ES waveforms that are retracked using algorithms and parametrization that can be found in literature. They basically use the same algorithm and parametrization for both data sets (CS2 and ES). They find significant

discrepancies between CS2 and Envisat. In particular, they obtain primarily negative freeboard and winter growth rates for ES, and positive freeboard and winter growth rates for CS2. The authors explain this by an dissimilar impact of ice surface roughness and snow volume scattering on SAR (CS2) and pulse-limited (ES) altimetry. Given this, they use the freeboard height difference between the two datasets as a function of the waveform pulse-peakiness to correct the ES freeboard to be aligned with the CS2 freeboard. They show the benefit of this bias correction by comparing CS2, ES and the corrected ES freeboard (ESC) with ice draft measurements from moorings in the Beaufort Gyre. The comparison reveals a good agreement between the in situ data and the CS2 ice draft, and a significant improvement, using the ESC data instead of the ES data set. Finally, the authors present a sea-ice thickness timeseries from 2002-2015, using ESC and CS2.

The paper gives attention to the problem of comparing sea-ice freeboard/thickness estimates that are obtained from different sensors. Though, both altimeters use Ku-band, the different footprints/technique (SAR vs. puls-limited) prevent uniform processing and parametrization, in particular with respect to the retracking of the waveforms using empirical threshold retrackers. In order to obtain consistent timeseries, bias corrections between different satellite eras are crucial. I think, the approach using the pulse-peakiness for the bias correction deserves publication. However, in the current form, I have some major concerns:

1. In general, I have the feeling that the paper lacks crucial information regarding the methodology, certainly the freeboard processing. Since you indicate using the TFMRA retracker, a very important information, which I could not find, concerns the retracker thresholds. Which values have been used here? Did the authors used the same for CS2 and ES (which I assume)? You refer to the ESA SI-CCI project, but without any reference. The reference Peacock and Laxon (2004) and Laxon et al. (2004) is acknowledged, but just gives a rough idea of the processing. Since you compare freeboard, this a key point of the study and needs much more detailed information. Here,

it would be also beneficial to show CS2 and ES waveforms with the corresponding re-tracking points. Also, I would suggest to include an orbit example, showing the along track ice surface elevations, sea surface height and detected leads. This would also highlight the differences between CS2 and ES (ESC).

2. Surely, the Envisat freeboard will be biased when using the same retracking parametrization as for CS2. But still, almost uniformly negative freeboard seems strange to me. But with the few details about processing given in the paper, it is hard to guess the reason.

3. I find the motivation and structure of the paper misleading as well as some terms that are used misleadingly ("negative freeboard", "surface diffusion"). As I understand, you process CS2 and ES freeboard using the same retracking algorithm and parametriza-tion. Then, you compare CS2 and ES, finding negative freeboard and winter growth rates for ES. For the reader, it seems that, a priori, you assume that you would get comparable results when applying the same method for ES as for CS2. Furthermore, CS2 freeboard might be biased as well, though less than ES, as the comparison with the in situ data indicates. Due to the different mode/footprint (SAR/pulse-limited), the effects of surface roughness and volume scattering are represented differently in the CS2 and ES radar echoes. Therefore, it seems evident that using the same threshold parametrization will lead to a more or less substantial bias in both data sets. I suggest to avoid using "negative freeboard" and "negative growth rates", since here, it is not a physical effect as in the Antarctic (flooded sea ice causes negative freeboard), but a bias due to the retracking parametrization. I would also recommend to revise the structure: Make clear that your motivation is to produce a consistent data set. Then, produce CS2/ES freeboard, using the same parametrization, but clarifying that differ-ences are expected. Then, only show the difference plots (CS2-ES), not the absolute freeboard necessarily (move Fig 2c to Fig 3 and discard Fig2 a/b ). Afterwards, you can introduce the correction function. You could add a figure then showing the abso-lute freeboard of CS2 and ESC (similar to former Fig 2 a/b) and the difference between

CS2 and ESC.

4. While I agree that Fig.7 is convincing and showing the entire time series is attracting, I think this also needs a more in-deep analysis and information. Over which area have you averaged? How did you deal with the pole holes? Also, separation between FYI and MYI would be interesting. And finally, uncertainty estimates are missing. I would consider discarding/changing this part and rather focus on the overlap years. I would like to see the sea-ice thickness distribution (monthly histograms) for CS2 and ESC for 2010-2012 and corresponding statistics.

In addition, I find sentences sometimes misleading and a bit unclear or too unspecific. I think this can be improved (see in the specific comments). As stated above, in my opinion, major revisions are needed before the paper can be considered for publication.

Specific Comments:

Title: no fullstop.

page 1:

l1: sea-ice . . . I suggest to use hyphenation here and in general, improves readability, though not used uniformly in literature.

l3: "as free of instrumental error as possible". . . this sounds a bit odd. And also, as stated above, I think the goal should rather be to produce consistent time series. Of course, reducing uncertainties is important as well, but doing this individually for both datasets does not guarantee a consistent time series. Any assumptions we have to make for the parametrization may introduce a bias in one of the data sets.

l4: . . . height(s)

l4-8: As mentioned in the general comments, the authors should avoid using "negative freeboard" and "negative winter growth rates". In particular for the abstract, this is very misleading.

[Figure]

l9-10: "Following..." In my opinion, this is the key message of the paper.

page 2:

l15-19: "While the...": As you mentioned, the SI-CCI product is a prototype product, which has not been published in a journal yet. I suggest to delete these two sentences as they do not really add value to the introduction.

l23: I have the feeling that the authors associate "bias" with "accuracy". While I agree that one can obtain more accurate freeboard and thickness estimates with CS2 (thanks to SAR altimetry), you seem to refer to the bias in the ES data. As mentioned above, this is a bias, which can be corrected (to some point, same as for CS2). It does not necessarily tell us something about the actual accuracy. And also, you argue that the bias in the Envisat ice thickness is driven by the freeboard and not by the freeboard-to-thickness conversion. Why should it be driven by the freeboard-to-thickness? Only, if you use different snow depth parametrization and other density values. Why should you?. I suggest to rephrase the paragraph and rather focus on the consistency between CS2 and Envisat.

page 3

l29: "than" = as l32: What does the CTOH netcdfs contain? geo-located waveforms? l1b elevations? What kind of data are you using? Please, be more specific here.

Section 2: Please be more specific: Which retracker thresholds have been used? It is true that the TFMRA is described already in Helm et al. (2014) (over land ice) and Ricker et al. (2014) (over sea ice). But a short description of the main processing steps is missing here from my point of view.

page 4

l12: "than" = as

l14: Which sea-level corrections do you mean here? DTU15? Tides?

l19-20: You refer to the SI-CCI project but without a reference. This is not very helpful for readers who are not involved in this project.

l21: In general, i suggest reducing the usage of "indeed".

l29-31: I agree that discarding these waveforms might lead to a bias. On the other hand, these waveforms can also result from off-nadir leads (mixed lead-ice waveform), similar shape as thin smooth nadir FYI, introducing a range bias.

l26: WF represents the echo power distribution, no?

page 5

l1: [upper] PP . . .. [lower] PP . . .

l10: "In Laxon . . ." . . . Are you sure? Didn't they use a Gaussian plus exponential model fit for lead waveforms?

l19: "the TFMRA retracker is parametrized identically" . . . Given that, it is seems clear that there will be a bias.

l21-22: Ricker et al. (2016): "The Impact of Geophysical Corrections on Sea-Ice Freeboard Retrieved from Satellite Altimetry" shows that for major parts of the Arctic, the geo-corrections (tides, wet/dry tropospheric Correction, etc.) do not really matter on basin scale. It is mostly the MSS playing a crucial role for the sea-level interpolation.

l24-25: Can the authors provide an along track plot for an orbit? With freeboard, ice/sea surface elevations, detected leads, and also including the filtered retrievals.

l24-26: Why do the authors use a 12.5 km grid (instead of 25 km for example)? Because in the following, you use a 100 km radius for the smoothing? Why such a large radius? I think you will loose lots of details in the spatial thickness/freeboard distribution, also the SARIN box seems to be "interpolated".

page 6

l11: "every" = any

l26: Which density are you using then? I cannot find a number.

l30: "An another" . . . typo

page7

l17: "The parameter . . ." I think it would be better to name it here already and then refer to section 3.3.

l17-18: Again, I find the spatial smoothing too coarse and certainly the SARIN box should be masked when not using the SARIN data.

l18-21: I do not really understand why the authors obtain such a freeboard (-13 cm in average). Even if you use the same threshold as for CS2, I would assume the freeboard to be mostly positive, see Schwegmann et al. (2016). It means that your lead elevations are significantly higher than those from the ice surface. Though I acknowledge that, in contrast to Schwegmann et al. (2016), the authors us the same retracker for both ES leads and ES sea-ice waveforms. Did you check for off-nadir leads? This could also be an issue. Again, I think more information about the freeboard processing are necessary here, for example showing lead fractions and an example for the along track processing.

page 8

section 3.2: I find this section misleading and not well understandable. What do you mean with "surface diffusion"? The Impact of surface roughness?

l32: "As suggested by the visual observation": rephrase, for example: "As suggested (indicated) by Fig.3"

l24-25: ". . .and/or melted snow" . . . Melted snow in November? I am not sure about that, at least not on basin scale. Moreover, this would mean that your observed freeboard is likely not ice freeboard anymore.

page 9

l3, Last paragraph: I do not really understand the point here. Do you mean the impact of surface roughness? Surely, this has an impact when using CS2 SAR altimetry on the one hand and ES pulse limited altimetry on the other hand. But again, I would argue that this is rather a retracking calibration/parametrization issue, when using a threshold retracker.

l15-17: The bias is also a question of how well the thresholds are calibrated. This counts for both CS2 and ES.

page 10

l3: "Looking at" -> "Considering"

section 3.4: So you first tune your ice thickness retrieval? Why are you using different densities here? Why not the same as for the freeboard-to-thickness conversion? This should be consistent. Moreover, you first tune your ice thickness and then you conclude that there is a good agreement with the mooring ice draft data. This is not surprising.

page 11

section 3.5: As suggested above, I think a more in-deep analysis is needed here if you want to keep this part. I would rather focus on the comparison during the overlap years.

page 12

The discussion is very short and overlaps with the conclusion section. Actually, the authors mixed "Results" and "Discussion" in the "Results" section. I suggest, either remove the "Discussion" section and call it (Results and Discussion) or separate them explicitly (which I would prefer).

Figure 3: Color tables: I find the usage of "polar" color tables confusing when they are not centered. May be, consider using a non-polar table, especially for PP, which is not a divergent data set.

Figure 4: I suggest to add CS2 waveforms and corresponding retracking points.

---

## Referee Comment (RC3) · T. W. K. Armitage (Referee) · 18 Feb 2017

The paper presents a new record of sea ice freeboard and thickness from CS2 and Envisat. A new method for levelling Envisat freeboard against CS2 based on PP is presented and represents an interesting and novel approach that could be very valuable for constructing longer time series' of freeboard/thickness. The authors evaluate their data against ice draft from the BGEP moorings and find good agreement, which lends credibility to their processing methodology. In particular, the agreement seen between Envisat and the moorings prior to the CS2 period is encouraging.

The paper is in general well-presented and the subject matter is certainly relevant for publication in The Cryosphere, however I believe some major revisions are required.

My concerns are broadly in line with the previous two reviewers; I believe that the authors' treatment of the Envisat data is not up to the standard of the current state of the art for radar altimeter sea ice processing. Whilst I have some strong criticisms of the methodology/interpretation, I have tried to provide a comprehensive review as I believe this paper deserves publication.

Major comments

1. My major concern with this paper is the interpretation that the difference between the Envisat and CS2 freeboard is due to a "dissimilar impact of ice roughness and snow volume scattering" (in the abstract, and throughout the manuscript). I prefer the interpretation that the difference (presented in figure 2a&b) is caused by the high sensitivity of the pulse-limited Envisat data to off nadir ranging as a result of the footprint size compared to CS2. Figure 3 shows that the high PP and highly biased Envisat freeboard is in areas where we might expect higher lead fractions, and that the PP is particularly high in November when there is rapid ice formation and open water areas. The assertion that the lower PP areas correspond to areas of MYI is not backed up by Figure 3b at all, in fact it shows high PP corresponding to the MIZ and polynya areas. In my opinion, the highly negative freeboard shown in Figure 2b (which cannot be published as is) is a direct result of the fact that the authors make use of waveforms with intermediate PP values. These waveforms will be highly contaminated by off nadir scattering, which causes the low sea ice elevation estimates, and hence negative freeboard when differenced with the local sea level. The authors need to improve their treatment of the Envisat data before it can be considered 'state of the art' and is suitable for publication. (See my specific comments below).

2. Related to this is the waveform interpretation. The authors assert that waveforms with intermediate PP values originate from thin level ice, however these waveforms are conventionally interpreted as showing 'mixed' scattering behavior. The 'conventional' interpretation is backed up by publications which compare altimeter returns with coincident imagery [e.g., Peacock & Laxon (2004), Armitage & Davison (2014)]. As

well as this, it is known that sea ice is rarely homogeneous at the scale of altimeter footprints (even SAR footprints), so you would almost always expect mixed scattering behavior to be present in echoes over sea ice. I believe that the waveforms presented in Figure 4 also show mixed scattering behavior – they all have a diffuse scattering component corresponding to the sea ice, and each one has a specular part super-imposed on top, presumably corresponding to leads or thin, freshly formed ice. You should plot the absolute power of the waveforms – is the diffuse scattering part of the waveforms remaining at a similar level, with different amount of specular scattering? I would require much more convincing, including detailed comparison with imagery, and possibly scatterometry (to show roughness), to be convinced by the interpretation that the intermediate waveforms correspond to thin, level ice.

3. The reference to "ice surface diffusion" and "surface diffusion variability" throughout the manuscript is confusing, and I do not know what the authors are actually referring to. I don't think I have come across this terminology in any other publications on satellite altimetry. You need to clarify, or adopt more conventional terminology. In some parts, it seems that you are implying that the different footprint shape/size changes the surface/volume scattering components of the ice (e.g., page 3, line 14-16). As far as I am aware, the surface/volume scattering depends on the frequency, the angle of incidence, and surface properties like grain size and water/salt content. I don't see how footprint size or shape can affect these properties?

4. I think it should be made clear throughout the manuscript that you are actually comparing the "radar freeboard" rather than "sea ice freeboard" e.g., page 1, line 5. This is particularly important when you're comparing the two instruments. For example, you say that the Envisat freeboard decreases during the season whilst CS2 increases – in actual fact the freeboard is independent of the altimeter (it is a geophysical quantity), but the radar freeboard that is retrieved by the altimeter can be different with different instruments. This distinction has been made in other publications (e.g., Ricker et al, 2014, Armitage and Ridout (2015)) and accounts for the fact that the altimeter freeboard may not correspond directly to the ice-snow interface.

5. Finally, I would consider splitting this paper into two. The first would concern the technical aspects of making a consistent sea ice thickness time series from two different altimeters, and evaluation of the data against in situ and airborne data. The second would use the decade+ long time series to do some science! The scientific value of this dataset is large, and it is wasted here – section 3.5 is just two paragraphs. If you retain the 'scientific' part of this manuscript, you should provide some interpretation – what is driving the inter-annual and long term changes of ice thickness? You should also provide maps of the sea ice thickness through the period, for example autumn (Oct&Nov) and spring (Feb&Mar) average thickness.

Specific comments:

Throughout the manuscript: the authors consistently refer to "freeboard height" - it is a personal preference but I think that you just need to say "freeboard", and not "freeboard height".

Page 1, line 3: "..free of instrumental error as possible". This is a rather trivial statement (of course you wish to minimize instrumental error) however it also misses the point that sea ice thickness uncertainty is dominated by snow loading error, not instrumental error.

Page 1, line 4: It's more accurate to say that you compared freeboard during the 2010/11 and 2011/12 sea ice growth seasons.

Page 1, line 10-12: It isn't valid to present a comparison of the EnvisatPP data with CS2 as a significant result because you are using CS2 to calibrate the EnvisatPP data – so the 'improvement' is by construction! The BGEP comparison is more significant.

Page 1, line 18-19. It would be interesting to test exactly how much ice volume Envisat is missing in the 'pole hole', by comparison with CS2 and ICESat. The 'circumpolar' claim (here and elsewhere in the manuscript) is arguable, due to the size of the Envisat
pole hole.

Page 2, line 9: "For *more* than a decade,..." or "Since 2003,..."

Page 2, line 14 and page 3, line 3-19: "LRM" – you should refer to the Envisat data as "pulse-limited" rather than "LRM". Low resolution mode is specific to CS2 and is just conventional pulse limited operation.

Page 2, line 22: Some references are missing: Ricker et al. (2014), Kurtz et al. (2014), Tilling et al. (2015).

Page 2, line 23-page 3, line 2: The "important question" discussed here is not a question at all: CS2 provides better estimates of ice thickness than Envisat because it was designed to! In the late 90s, the question was asked, how can we improve altimeter design to better capture interannual and seasonal sea ice thickness variability? The answer was CS2 – a SAR altimeter, with very high inclination orbit.

Page 2, line 25-26: The freeboard to thickness conversion uncertainty affects both Envisat and CS2 in the same way, so would not result in a bias in Envisat.

Page 4, line 12: the bandwidth (receive) of SIRAL is the same as Envisat, not similar.

Page 4 line 27-page 5, line 7: This relates to my major comment above. You need to provide substantial evidence that intermediate PP waveforms "likely result from thin and relatively flat sea ice", as this would be contrary to the current understanding as presented in the literature. You say that filtering these data may bias the sea ice thickness high, however there is no evidence of this in other publications presenting comparisons with in situ data (e.g., Tilling et al (2015)). In fact, including these waveforms produces the extreme negative freeboard maps present in Figure 2b. For me, you would have to develop and demonstrate an extremely robust retracker to make use of intermediate PP Envisat waveforms.

Section 2.3: It is surely not valid to use the exact same processing for Envisat and CS2 (PP thresholds, retracker parameters) given the fundamental difference between the

instruments??

Page 5, line 10-12: Two different retrackers are used in Laxon et al (2013), hence the need for the bias correction. As a point of reference, the SICCI ATBD is actually based on the CPOM processing presented by Laxon et al (2013).

Page 5, line 16-19: Has this retracker been demonstrated for Envisat, or just CS2? If not, then you need to do a proper assessment on the Envisat data.

Page 5, line 21-27: Sea level interpolation causes errors because of lack of lead tie points, snagging, or use of a poor geoid/MSS model. Geophysical corrections have a much smaller effect, as I think another reviewer pointed out. Your method for treating sea level interpolation is new and needs to be demonstrated more robustly against current algorithms.

Page 6, line 4-6: I believe it was Laxon et al. (2013) who first used the "Warren/50% on FYI" methodology, not Kwok & Cunningham (2015).

Page 6/Figure 1: monthly snow depth – wouldn't it be better to use daily ice type masks and match to individual altimeter orbits? The location/size of the MYI area can vary quite a lot over the course of a month.

Page 7, line 17, Figure 2c: You should introduce figure 2c here or move it – perhaps move it to Figure 3.

Figure 3: I find the colourbar used for Figure 3 misleading – normally the red-blue "polar" colourbar is centred on zero, to show positive/negative values. It also makes it appear as though the PP is zero in large areas.

Page 8, line 4-5: Here is an example of misleading use of "thicker freeboard". The radar freeboards are different, the ice freeboard stays the same.

Section 3.2: This section will need considerable revision based on my major comments.

Page 9, lines 11-18: Is the first part of this paragraph necessary? Consider cutting.

Section 3.3 is good, the most interesting/important development of the paper.

Page 10, line 4-5, Figure 18a,b,j,k. I think it's worth noting that the CS2/EnvisatPP are so similar *by construction*. Currently the paper makes is appear like the agreement between EnvisatPP and CS2 is a significant result in itself, but it is simply a consequence of levelling the CS2 against the Envisat data. This doesn't detract from low RMSE or the good agreement seen with the BGEP moorings, but is an important point.

Section 3.4: I wonder if you could do your evaluation with any other datasets? E.g., Fram Strait moorings have been in place for a long time, Operation IceBridge goes back to 2009, EM-bird data.

Section 3.5: I think this section should be greatly expanded, or else written up as a separate paper. What is driving interannual to decadal thickness variability? This can be done by comparison with ice drift, temperature records, climate indices (e.g., AO). You should compare the Envisat thickness with ICESat. You should present seasonal maps of ice thickness for the entire time period. Are changes in basin mean thickness reflected in changes in volume? What are the implication for heat/freshwater storage?

Page 11, line 23: The references should be in chronological

---

## Author Comment (AC1) · 11 Apr 2017

First of all, we would like to thank all three reviewers as well as the Editor for their constructive comments and advices that truly helped to improve the first version of our manuscript.

The response to the reviewers is developed as follows:

-The first section provides general comments on the changes and reviews.
-The second part is a detailed answer to each reviewer.
-The last part is a summary of all changes operated in the new version.

I-General comments and modifications:

+ About the freeboard height retrieval

The freeboard height methodology is now further detailed in the new version of the manuscript. In particular, a new section with an along-track analysis is now provided and the retrieval steps are further discussed. We also combine optical imagery with radar altimeter measurement to improve the flow/lead detection and we make the appropriate changes in the freeboard height retrievals.

+ About the Envisat freeboard estimates

First of all, we would like to remind the reviewers that this manuscript would potentially be the first study showing Envisat circumpolar Arctic freeboard maps. In previous published studies, only ice thickness maps were presented and we therefore have no other published study on this topic to rely on.
Regarding the negative Envisat freeboard estimates: as this effect was already described and corrected in sea-ice studies (Giles et al., 2008, Laxon et al., 2013) and ocean studies (Giles et al., 2012, Armitage et al., 2017) we thought that it was not necessary to spend too much time on this topic. Considering the reviewers comments, we now give more insights and explanations on this phenomenon. In particular, the along-track analysis section should truly helps to understand the negative freeboard estimates obtained with Envisat.
Regarding the spatial variability of the native Envisat freeboard estimates: the 2010-2012 period is unfortunately not a good period to observe a high variability of radar freeboard height as the MYI fraction is very low. Having said that, if you look at our estimates for let's say March 2007 (see bellow) you will see that the native Envisat freeboard estimates still capture some coherent spatial variability despite the negative freeboard estimates.

+ About the structure of the manuscript

Following reviewers comments, the structure of the manuscript was modified in order to highlight more clearly the goal of the study: improving Envisat freeboard retrievals in the aim of producing accurate Arctic ice thickness estimates.
In addition to the extra section concerning the along-track analysis, we decided to follow the reviewers comments and to remove the time-series section. These results will be further developed in a new study.

[Figure]

*Figure 1: Envisat "native" radar freeboard for March 2007.*

II-Detailed answer to referee #1:

**1. The Envisat FB product before PP correction Looking at figure 2 middle column, one can see that the Envisat freeboards are unrealistic. For one, they are negative – something that the authors just attribute to "the difference of ice surface characteristics between leads and ice floes as well as the use of a threshold retracker drive a large bias on the estimation of Envisat freeboard height". I am confident that the culprit is elsewhere.**

As it is now further explained in the new version of the manuscript, most freeboard studies (Laxon et al., 1994; Giles et al., 2008; Laxon et al., 2013) and sea level studies (Giles et al., 2012; Armitage et al., 2017) artificially correct the bias due to the difference of specularity between rough ice and leads or rough ocean and leads by using 2 different retracking algorithms. The physical origin of this bias was certainly not enough detailed in the first version of the manuscript. In the new version we try to give more insights about this phenomenon.

**We've tested the TFMRA retracking scheme for Envisat as well in the CCI project, and we've arrived at more or less similar looking freeboard maps as with the original CCI retracking scheme. We seem to be missing the thinnest and the thickest ice, but freeboards are positive as they are supposed to be and the thickness pattern reflects reality (even with the thinnest and thickest ice missing). And furthermore, we do not see very high freeboards in the marginal ice zone.**

Considering the current literature and our personal experience, we would be quite surprised that positive sea-ice freeboard estimates can be retrieved with a single threshold retracker (TMFRA in that case) and without applying any further correction. We haven't found such results on line but if you provide us with a dataset or with published results, we would be happy to compare both datasets.

**I try to be a good reviewer and speculate possible causes for the Envisat freeboards being much off. My guess is that this may be due to off-nadir leads or new ice dominating significant number of waveforms. The authors give very little notice to filtering out mixed waveforms. Or filtering in general – it is hardly mentioned anywhere in the paper. They argue that they should keep in the waveforms with intermediate PP since they represent thin and undeformed ice. Fair enough, but at the same time they are letting in a lot of waveforms with deformed ice in the nadir and flat areas off-nadir which will lead into the retracker catching the off-nadir rise and biasing the elevation estimate low. This is consistent with the lowest freeboards seen in the area with lot of deformed ice (there will always be a significant number of flat new ice or leads around). This is less of a problem in the area of new ice near the margins, where the ice is more or less flat all around and in likelihood there is a specular surface in the nadir as well. All consistent with the pattern in figure 2. The authors hint that the use of TFMRA retracker is robust for off nadir reflections (page 5, lines 16-19). That is somewhat true, but it does not remove the need to filter out dubious waveforms – even Helm et al 2014 that the authors cite for the TFMRA have a filtering scheme to remove "bad waveforms" before retracking. I suggest the authors build one too and check if that improves their not-PP corrected freeboards. The SI-CCI scheme most likely filters too much waveforms, but I would still argue that some kind of filtering is required. Finally , much less likely culprit than previous one , but worth mentioning still since applying an inverted snow correction (that is, a bug in code) results into something bit like the maps in Figure 2. The main reason I'm mentioning this is that I once had that bug in my code and the Figure 2 reminds me much of it. Don't waste too much time on this, but do check your snow propagation correction code.**

In the new version of the manuscript, we use optical imagery to identify PP observations for which the waveform echoes are likely biased by mixed surfaces (leads+floes). Based on this new meethodology, the CryoSat-2 and Envisat freeboard is-recalculated. As a result, we observe that the freeboard is somehow improved on MYI. However, despite the use of this filter, the radar freeboard remains quite negative.

**2. Theoretical justification of the PP correction The manuscript fails to explain the theoretical background of why exactly small PP (or more diffuse waveforms or heavily deformed ice) results into retracker picking up the tracking point later in the waveform that it would if the waveform was more peaky (less diffuse and most likely originating from less deformed ice). The authors state that "ice surface diffusion has a higher impact on LRM altimeters" but this needs to be backed up by something solid because from the evidence authors give. Because of the unrealistic Envisat FB, I do not believe that the disagreement of the Envisat and CS-2 freeboards is mainly due to surface diffusion. If the authors do not, theoretically step by step, explain the process of ice surface diffusion impacting LRM altimeter estimates, a good referee could (and should) claim that it is just as likely that what we are seeing here instead is something profoundly wrong with the Envisat FB retrieval and that something is connected to pulse peakiness.**

In the new version, we try to further explain how the surface specularity/diffusion acts on the LRM radar signal and why it impacts the freeboard height retrieval. In particular, we add a section on waveforms shape and on-track freeboard retrievals.
In all sea-ice studies, the PP is used as a proxy of surface diffusion/specularity to identify leads and ice floes. It is therefore the most relevant parameter according to the literature to be used as a proxy for surface roughness. In particular, it has been shown in the study by Zygmuntovska et al. (2013) that the PP is a fairly good proxy to distinguish rough MYI from specular FYI.

**The y(PP) is problematic anyway. Naturally, applying any correction derived from the difference of the two freeboard datasets will make the two agree. Strongest point the authors give for the use of the y(PP) correction is the improvement it brings to the fit of BGEP data throughout the Envisat period. This is all good and well, but looking at figure 6, the only real improvement is the level correction of about $1 - 1,5$ m to the (unrealistic and often even negative) Envisat draft estimates. I would argue that what we see here is the constant term of y(PP) – there must be one since the dashed line in figure 5 does not cross zero – just fixes the large negative bias that the somehow broken Envisat freeboard method produces.**

Figure 1 (in this document) shows a native Envisat freeboard map. As mentioned earlier, this map clearly displays coherent spatial variations with thicker values over MYI and thinner values of FYI as shown over Antarctic sea-ice in the study by Schwegmann et al. (2015). The native Envisat freeboard estimates bring therefore essential informations for the final estimates without which the corrected Envisat estimates would be highly inaccurate.
Clearly, the figure showing comparison with the BGEP moorings shows that the most important correction is the sea-level one (constant correction). However, the large improvement in the correlation coefficient is only due to correction of the bias driven by the variability of specularity of ice floes (the correlation coefficient does not depend on any potential constant bias). In order to further highlight this improvement we now show both the Envisat and Envisat/PP correlation coeffcient, average bias and RMSD in a table.

**After the harsh critique above, I should mention that the idea presented in the manuscript is**

**most definitely on the right track! A PP based correction would improve the problems of Envisat FB retrieval drastically. I know of similar attempts in the altimetric community lately. After fixing their uncorrected FB estimates and giving a theoretical justification of how the correction works, this will be a really good paper and I commend the authors for coming up with the idea and publishing it first. Problem with the manuscript at the moment is, that even if the final result of corrected Envisat freeboards seems to comply with validation data, the paper fails to give rigorous explanation exactly what are the processes why their methodology works.**

Thanks for your encouragements. Your comments truly helped to identify sections that needed to be clarified. We hope that the modifications in the freeboard height retrievals as well as the new explanations provided will be more convincing for any potential reader.

**1. Pan-arctic claim**

**The authors claim that they have created a pan-Arctic thickness estimate. They have not, since they have excluded all of the Arctic above 81,5 N. Thus I recommend the authors follow the lead of Giles et al and stick with "circumpolar" (or something similar) to emphasise that their estimate does not cover all of the Arctic.**

That's right. While the last section has been removed, we will stick to "circumpolar" instead of Pan-Arctic for now and in our future studies.

**2. TFMRA parameters**

**Nowhere in the paper the authors state, which threshold value they use for the TFMRA. 50%? It should be mentioned. Like other TFMRA parameters as well.**

50% indeed. More details are now provided in the new version of the manuscript.

**3. Local sea level interpolation**

**The description of the sea level interpolation is thin (page 5, lines 20 – 24). Not sure if the interpolation of leads could contribute to the unrealistic negative freeboards, but it is worth checking. Nevertheless, the authors must include a better description of the sea level interpolation – how exactly is it done? Taking a mean of all lead elevations within 25 km or some kind of along-track interpolation?**

This section was slightly modified to be more clear. Basically, for each 25 km segment we check if there is a lead. If not, no freeboard is estimated. If they are leads, the freeboard is estimated as the difference between the level of floes and the average level of leads.

**4. PP correction – are leads included?**

**On page 5, line 27 it is stated that PP is also averaged into gridded maps. Does this include the waveforms that are classified as leads?**

No, only ice floes echoes are kept to construct the gridded PP fields. It is now clearly stated in the manuscript.

**If it does, this will have a consequence to the PP correction – that is, areas with lot of leads will eventually have a stronger correction in the direction of thinner ice.**

That is correct. It is important here to highlight an interesting phenomenon: usually leads are associated with off-nadir reflections. However, in our study, the regions with a high PP (potentially characterized by a high density of leads) are the regions with the lower underestimation of surface elevation. This result suggests therefore that leads have the same impact than specular sea-ice: they tend to decrease the size of the effective radar footprint making waveform echoes sharper and reducing the altimetric range (when using an empirical threshold retracker.

**5. Mathematical description of y(PP)**

**The authors really must give a more thorough description of the y(PP). Is the y(PP) constant throughout the winter? I reckon it is the black dashed line in Figure 5, and constant over time**

**and place and calculated on the gridded level and not for individual measurements, but a mathematical formulation would be most welcome.**

We now provide with a mathematical description of y(PP) so anyone can reproduce our results. We do keep a constant y(PP) during winter and we show that the Envisat/PP radar freeboard is relatively similar (low RMSD)  as CryoSat-2 during all months of the period of study.

III-Summary of changes #1:

With respect to the new version manuscript order:

→ The abstract and introduction have been slightly re-written to clearly express the aim of this study and the key steps.

→ The freeboard processing is now more detailed (sea level, TFMRA retracker, etc). In addition, we add a comparison with Landsat images to validate the use of our PP thresholds.

→ Changes in the freeboard processing chains were applied, all freeboard estimates were re-calculated and figures were updated.

→ The ice density parametrization has been modified and is now more in phase with the literature (882 kg/m³ for MYI and 917 kg/m³).

→ A short analysis of CryoSat-2 and Envisat waveforms is now provided (sect 3.1)

→ An analysis of along-track radar freeboard is now provided (section 3.2).

→ Section 3.3 and 3.4 have been inverted.

→ The section showing ice thickness time series has been removed and will be part of a future study.

→ Tables with statistical parameters were improved

→ In general, the physical impact of ice surface properties on the radar signal is more clearly explained.

---

## Author Comment (AC2) · 11 Apr 2017

First of all, we would like to thank all three reviewers as well as the Editor for their constructive comments and advices that truly helped to improve the first version of our manuscript.

The response to the reviewers is developed as follows:

-The first section provides general comments on the changes and reviews.
-The second part is a detailed answer to each reviewer.
-The last part is a summary of all changes operated in the new version.

I-General comments and modifications:

+ About the freeboard height retrieval

    The freeboard height methodology is now further detailed in the new version of the manuscript. In particular, a new section with an along-track analysis is now provided and the retrieval steps are further discussed. We also combine optical imagery with radar altimeter measurement to improve the flow/lead detection and we make the appropriate changes in the freeboard height retrievals.

+ About the Envisat freeboard estimates

    First of all, we would like to remind the reviewers that this manuscript would potentially be the first study showing Envisat circumpolar Arctic freeboard maps. In previous published studies, only ice thickness maps were presented and we therefore have no other published study on this topic to rely on.
    Regarding the negative Envisat freeboard estimates: as this effect was already described and corrected in sea-ice studies (Giles et al., 2008, Laxon et al., 2013) and ocean studies (Giles et al., 2012, Armitage et al., 2017) we thought that it was not necessary to spend too much time on this topic. Considering the reviewers comments, we now give more insights and explanations on this phenomenon. In particular, the along-track analysis section should truly helps to understand the negative freeboard estimates obtained with Envisat.
    Regarding the spatial variability of the native Envisat freeboard estimates: the 2010-2012 period is unfortunately not a good period to observe a high variability of radar freeboard height as the MYI fraction is very low. Having said that, if you look at our estimates for let's say March 2007 (see bellow) you will see that the native Envisat freeboard estimates still capture some coherent spatial variability despite the negative freeboard estimates.

+ About the structure of the manuscript

    Following reviewers comments, the structure of the manuscript was modified in order to highlight more clearly the goal of the study: improving Envisat freeboard retrievals in the aim of producing accurate Arctic ice thickness estimates.
In addition to the extra section concerning the along-track analysis, we decided to follow the reviewers comments and to remove the time-series section. These results will be further developed in a new study.

[Figure]

*Figure 1: Envisat "native" radar freeboard for March 2007.*

Detailed answer to referee #2:

**1. In general, I have the feeling that the paper lacks crucial information regarding the methodology, certainly the freeboard processing. Since you indicate using the TFMRA retracker, a very important information, which I could not find, concerns the retracker thresholds. Which values have been used here? Did the authors used the same for CS2 and ES (which I assume)?**

These informations are now provided in the new version of the manuscript including the description of the TFMRA retracker. It is now more clearly stated that the same retracker is used for CryoSat-2 and Envisat as well as for leads and ice floes.

**You refer to the ESA SI-CCI project, but without any reference. The reference Peacock and Laxon (2004) and Laxon et al. (2004) is acknowledged, but just gives a rough idea of the processing. Since you compare freeboard, this a key point of the study and needs much more detailed information. Here, it would be also beneficial to show CS2 and ES waveforms with the corresponding retracking points. Also, I would suggest to include an orbit example, showing the along track ice surface elevations, sea surface height and detected leads. This would also highlight the differences between CS2 and ES (ESC).**

The section in which the freeboard retrieval is described has been improved. In particular, a comparison of radar observations with optical imagery is now provided and a section describing along-track freeboard estimates has been added as you recommended it.

**2. Surely, the Envisat freeboard will be biased when using the same retracking parametrization as for CS2. But still, almost uniformly negative freeboard seems strange to me. But with the few details about processing given in the paper, it is hard to guess the reason.**

Hopefully, the new sections and further explanations will clarify this particular phenomenon.

**3. I find the motivation and structure of the paper misleading as well as some terms that are used misleadingly ("negative freeboard", "surface diffusion"). As I understand, you process CS2 and ES freeboard using the same retracking algorithm and parametrization. Then, you compare CS2 and ES, finding negative freeboard and winter growth rates for ES. For the reader, it seems that, a priori, you assume that you would get comparable results when applying the same method for ES as for CS2.**

Excellent point. We clearly do not expect to obtain the same freeboard at the end of the processing chain. All we want to do is to minimize the impact of the processing chain on freeboard height differences between the two sensors. This is now stated in the manuscript.

**Furthermore, CS2 freeboard might be biased as well, though less than ES, as the comparison with the in situ data indicates. Due to the different mode/footprint (SAR/pulse-limited), the effects of surface roughness and volume scattering are represented differently in the CS2 and ES radar echoes. Therefore, it seems evident that using the same threshold parametrization will lead to a more or less substantial bias in both data sets. I suggest to avoid using "negative freeboard" and "negative growth rates", since here, it is not a physical effect as in the Antarctic (flooded sea ice causes negative freeboard), but a bias due to the retracking parametrization.**

In the new version of the manuscript, we do not use any longer these ambiguous terms.

**I would also recommend to revise the structure: Make clear that your motivation is to produce a consistent data set. Then, produce CS2/ES freeboard, using the same**

**parametrization, but clarifying that differences are expected. Then, only show the difference plots (CS2-ES), not the absolute freeboard necessarily (move Fig 2c to Fig 3 and discard Fig2 a/b ). Afterwards, you can introduce the correction function. You could add a figure then showing the absolute freeboard of CS2 and ESC (similar to former Fig 2 a/b) and the difference between CS2 and ESC.**

This is one of the major changes we operated. Thanks to your comments and suggestions, the "Result" section is now developed as follows:

       -Comparison of CryoSat-2 and Envisat waveform echoes

       -Along-track analysis of surface elevation and freeboard height

       -Gridded radar freeboard difference and link with ice surface properties

       -Improvement of the native Envisat freeboard height fields with the PP-correction

       -Validation of the approach with moorings observations

**4. While I agree that Fig.7 is convincing and showing the entire time series is attracting, I think this also needs a more in-deep analysis and information. Over which area have you averaged? How did you deal with the pole holes? Also, separation between FYI and MYI would be interesting. And finally, uncertainty estimates are missing. I would consider discarding/changing this part and rather focus on the overlap years. I would like to see the sea-ice thickness distribution (monthly histograms) for CS2 and ESC for 2010-2012 and corresponding statistics.**

This comment was also expressed by the other reviewers. We therefore chose to remove this section from the manuscript and to provide more complete results and explanations in a future study.

**Title: no fullstop.**

Ok

**page 1:**

**l1: sea-ice . . . I suggest to use hyphenation here and in general, improves readability, though not used uniformly in literature.**

Thanks for the advice. The hyphenation is now used in the manuscript.

**l3: "as free of instrumental error as possible". . . this sounds a bit odd. And also, as stated above, I think the goal should rather be to produce consistent time series. Of course, reducing uncertainties is important as well, but doing this individually for both datasets does not guarantee a consistent time series. Any assumptions we have to make for the parametrization may introduce a bias in one of the data sets.**

The abstract has been modified in order to clearly display the aim of this study: improving Envisat freeboard retrievals in the aim of producing accurate Arctic ice thickness estimates.

**l4: . . . height(s)**

This sentence was rephrased.

**l4-8: As mentioned in the general comments, the authors should avoid using "negative freeboard" and "negative winter growth rates". In particular for the abstract, this is very misleading.**

In the new version of the manuscript, "negative winter growth" is no longer employed and "negative freeboard" is used as few as possible.

**l9-10: "Following. . ." In my opinion, this is the key message of the paper.**

**page 2: l15-19: "While the. . .": As you mentioned, the SI-CCI product is a prototype product, which has not been published in a journal yet. I suggest to delete these two sentences as they do not really add value to the introduction.**

In fact, we do quote a published paper to refer to the SI-CCI product [Ridout and Tonboe, 2012]. As the SI-CCI product is the only previous study we can refer to, we would prefer to keep these sentences.

**l23: I have the feeling that the authors associate "bias" with "accuracy". While I agree that one can obtain more accurate freeboard and thickness estimates with CS2 (thanks to SAR altimetry), you seem to refer to the bias in the ES data. As mentioned above, this is a bias, which can be corrected (to some point, same as for CS2). It does not necessarily tell us something about the actual accuracy.**

Right. We applied corrections here and throughout the manuscript to take this comment into account.

**And also, you argue that the bias in the Envisat ice thickness is driven by the freeboard and not by the freeboardto-thickness conversion. Why should it be driven by the freeboard-to-thickness? Only, if you use different snow depth parametrization and other density values. Why should you?. I suggest to rephrase the paragraph and rather focus on the consistency between CS2 and Envisat.**

That is exactly the message we wanted to pass through. Hopefully, the changes we made will help to clarify the message.

**page 3**

**l29: "than" = as**

OK

**l32: What does the CTOH netcdfs contain? geo-located waveforms? l1b elevations? What kind of data are you using? Please, be more specific here.**

Even though more details are now provided in the new version, I am not sure what you mean by "what kind of data are you using" considering the information we already provide. Is it better now? If not, could you be more specific on you expectations please?

**Section 2: Please be more specific: Which retracker thresholds have been used? It is true that the TFMRA is described already in Helm et al. (2014) (over land ice) and Ricker et al. (2014) (over sea ice). But a short description of the main processing steps is missing here from my point of view.**

Right, a short description has been added.

**Page 4**
**l12: "than" = as**
OK
**l14: Which sea-level corrections do you mean here? DTU15? Tides?**
OK

**l19-20: You refer to the SI-CCI project but without a reference. This is not very helpful for readers who are not involved in this project.**

As explained above, we do give a reference for the SI-CCI project. In the new version, we repeat a few times this reference to help the reader.

**l21: In general, i suggest reducing the usage of "indeed".**

Changes were operated as much as possible.

**l29-31: I agree that discarding these waveforms might lead to a bias. On the other hand, these waveforms can also result from off-nadir leads (mixed lead-ice waveform), similar shape as thin smooth nadir FYI, introducing a range bias.**
As you and the other reviewers expressed the same concerns about the data filtering, we now filter our data to eliminate ambiguous observations that could potentially drive off-Nadir reflections. Further details on the filtering are provided in section 2.4.

**l26: WF represents the echo power distribution, no?**
Yes, it is now stated in the manuscript.

**page 5**
**l1: [upper] PP . . .. [lower] PP . . .**
This section has been deeply modified.

**l10: "In Laxon . . ." . . . Are you sure? Didn't they use a Gaussian plus exponential model fit for lead waveforms?**
You are absolutely right. This is now corrected.

**l19: "the TFMRA retracker is parametrized identically" . . . Given that, it is seems clear that there will be a bias.**
Definitely, yes. But this bias should be constant except if the sea-ice surface scattering has a different impact on one of the two sensors...

**l21-22: Ricker et al. (2016): "The Impact of Geophysical Corrections on Sea-Ice Freeboard Retrieved from Satellite Altimetry" shows that for major parts of the Arctic, the geo-corrections (tides, wet/dry tropospheric Correction, etc.) do not really matter on basin scale. It is mostly the MSS playing a crucial role for the sea-level interpolation.**
In areas where the lead density is relatively low and where the average between 2 leads gets larger, it is likely that even though the effects of these corrections is low, they are not negligible.

**l24-25: Can the authors provide an along track plot for an orbit? With freeboard, ice/sea surface elevations, detected leads, and also including the filtered retrievals.**
YES! We now provide such figure (within a brand new section). We hope it clarifies the explanation about the unrealistic Envisat freeboard values.

**l24-26: Why do the authors use a 12.5 km grid (instead of 25 km for example)? Because in the following, you use a 100 km radius for the smoothing?**
As a matter of fact, we simply took the same grid as the one used in the NSIDC sea-ice extent product.
**Why such a large radius? I think you will loose lots of details in the spatial thickness/freeboard distribution, also the SARIN box seems to be "interpolated".**
While we could use a lower radius for CryoSat-2 radar freeboard, the Envisat radar freeboard is much noisier and requires a wider smoothing.

**Page 6**
**l11: "every" = any**
OK

**l26: Which density are you using then? I cannot find a number.**
OK

**l30: "An another" . . . typo**
OK

**page7**
**l17: "The parameter . . ." I think it would be better to name it here already and then refer to section 3.3.**
This section was rephrased

**l17-18: Again, I find the spatial smoothing too coarse and certainly the SARIN box should be masked when not using the SARIN data.**
In the first maps we plotted, we filtered the data found in the SARIN box that were recovered by the coarse spatial smoothing. Unfortunately, the amount of thick MYI is quite rare during the period of study (2010-2012) and this filtering caused the loss of most of the thick freeboard estimates and made the y(PP) relation less valuable. This is the reason why we decided to keep these observations.

**l18-21: I do not really understand why the authors obtain such a freeboard (-13 cm in average). Even if you use the same threshold as for CS2, I would assume the freeboard to be mostly positive, see Schwegmann et al. (2015).**
You're right. But in Schwegmann et al. (2015), the authors used 2 retrackers, which artificially correct the negative bias. Everybody processes the LRM freeboard (Envisat, ERS, AltiKa) this way but the reason why such processing is applied is rarely discussed.

**It means that your lead elevations are significantly higher than those from the ice surface.**
As shown in section 3.2, yes indeed.

**Though I acknowledge that, in contrast to Schwegmann et al. (2016), the authors us the same retracker for both ES leads and ES sea-ice waveforms. Did you check for off-nadir leads? This could also be an issue. Again, I think more information about the freeboard processing are necessary here, for example showing lead fractions and an example for the along track processing.**
Hopefully, we now provide enough details to clarify this topic.

**page 8**
**section 3.2: I find this section misleading and not well understandable. What do you mean with "surface diffusion"? The Impact of surface roughness?**
Here and throughout the manuscript, we modified the way of explaining this phenomenon. In particular, we now describe the impact of the ice surface properties on the waveform shape and the consequences when using a threshold retracker.

**l32: "As suggested by the visual observation": rephrase, for example: "As suggested (indicated) by Fig.3"**
OK
**l24-25: ". . .and/or melted snow" . . . Melted snow in November? I am not sure about that, at least not on basin scale. Moreover, this would mean that your observed freeboard is likely not ice freeboard anymore.**
Right, we removed this part of the sentence...

**Page 9**
**l3, Last paragraph: I do not really understand the point here. Do you mean the impact of surface roughness? Surely, this has an impact when using CS2 SAR altimetry on the one hand**

**and ES pulse limited altimetry on the other hand. But again, I would argue that this is rather a retracking calibration/parametrization issue, when using a threshold retracker.**
**l15-17: The bias is also a question of how well the thresholds are calibrated. This counts for both CS2 and ES.**

The new sections provided in the second version should help to clarify this point.

**page 10**
**l3: "Looking at" -> "Considering"**
OK

**section 3.4: So you first tune your ice thickness retrieval? Why are you using different densities here? Why not the same as for the freeboard-to-thickness conversion? This should be consistent. Moreover, you first tune your ice thickness and then you conclude that there is a good agreement with the mooring ice draft data. This is not surprising.**
We now use the classical parametrization (882 kg/m³ for MYI and 917 kg/m³ for FYI).

**page 11 section 3.5: As suggested above, I think a more in-deep analysis is needed here if you want to keep this part. I would rather focus on the comparison during the overlap years.**
This section has been removed and will be part of a future study.

**page 12 The discussion is very short and overlaps with the conclusion section. Actually, the authors mixed "Results" and "Discussion" in the "Results" section. I suggest, either remove the "Discussion" section and call it (Results and Discussion) or separate them explicitly (which I would prefer).**
We tried to clearly distinguish the discussion and the conclusion section in the new version of the manuscript.

**Figure 3: Color tables: I find the usage of "polar" color tables confusing when they are not centered. May be, consider using a non-polar table, especially for PP, which is not a divergent data set.**
We changed back to "jet" in the new version.

III-Summary of changes #1:

With respect to the new version manuscript order:

→ The abstract and introduction have been slightly re-written to clearly express the aim of this study and the key steps.

→ The freeboard processing is now more detailed (sea level, TFMRA retracker, etc). In addition, we add a comparison with Landsat images to validate the use of our PP thresholds.

→ Changes in the freeboard processing chains were applied, all freeboard estimates were re-calculated and figures were updated.

→ The ice density parametrization has been modified and is now more in phase with the literature (882 kg/m³ for MYI and 917 kg/m³).

→ A short analysis of CryoSat-2 and Envisat waveforms is now provided (sect 3.1)

→ An analysis of along-track radar freeboard is now provided (section 3.2).

→ Section 3.3 and 3.4 have been inverted.

→ The section showing ice thickness time series has been removed and will be part of a future study.

→ Tables with statistical parameters were improved

→ In general, the physical impact of ice surface properties on the radar signal is more clearly explained.

---

## Author Comment (AC3) · 11 Apr 2017

First of all, we would like to thank all three reviewers as well as the Editor for their constructive comments and advices that truly helped to improve the first version of our manuscript.

The response to the reviewers is developed as follows:

-The first section provides general comments on the changes and reviews.

-The second part is a detailed answer to each reviewer.

-The last part is a summary of all changes operated in the new version.

**I-General comments and modifications:**

**+ About the freeboard height retrieval**

The freeboard height methodology is now further detailed in the new version of the manuscript. In particular, a new section with an along-track analysis is now provided and the retrieval steps are further discussed. We also combine optical imagery with radar altimeter measurement to improve the flow/lead detection and we make the appropriate changes in the freeboard height retrievals.

**+ About the Envisat freeboard estimates**

First of all, we would like to remind the reviewers that this manuscript would potentially be the first study showing Envisat circumpolar Arctic freeboard maps. In previous published studies, only ice thickness maps were presented and we therefore have no other published study on this topic to rely on.

Regarding the negative Envisat freeboard estimates: as this effect was already described and corrected in sea-ice studies (Giles et al., 2008, Laxon et al., 2013) and ocean studies (Giles et al., 2012, Armitage et al., 2017) we thought that it was not necessary to spend too much time on this topic. Considering the reviewers comments, we now give more insights and explanations on this phenomenon. In particular, the along-track analysis section should truly helps to understand the negative freeboard estimates obtained with Envisat.

Regarding the spatial variability of the native Envisat freeboard estimates: the 2010-2012 period is unfortunately not a good period to observe a high variability of radar freeboard height as the MYI fraction is very low. Having said that, if you look at our estimates for let's say March 2007 (see bellow) you will see that the native Envisat freeboard estimates still capture some coherent spatial variability despite the negative freeboard estimates.

**+ About the structure of the manuscript**

Following reviewers comments, the structure of the manuscript was modified in order to highlight more clearly the goal of the study: improving Envisat freeboard retrievals in the aim of producing accurate Arctic ice thickness estimates.

In addition to the extra section concerning the along-track analysis, we decided to follow the reviewers comments and to remove the time-series section. These results will be further developed

in a new study.

Figure 1: Envisat "native" radar freeboard for March 2007.

**Detailed answer to referee #3:**

1. My major concern with this paper is the interpretation that the difference between the Envisat and CS2 freeboard is due to a "dissimilar impact of ice roughness and snow volume scattering" (in the abstract, and throughout the manuscript). I prefer the interpretation that the difference (presented in figure 2a&b) is caused by the high sensitivity of the pulse-limited Envisat data to off nadir ranging as a result of the footprint size compared to CS2. Figure 3 shows that the high PP and highly biased Envisat freeboard is in areas where we might expect higher lead fractions, and that the PP is particularly high in November when there is rapid ice formation and open water areas. The assertion that the lower PP areas correspond to areas of MYI is not backed up by Figure 3b at all, in fact it shows high PP corresponding to the MIZ and polynya areas. In my opinion, the highly negative freeboard shown in Figure 2b (which cannot be published as is) is a direct result of the fact that the authors make use of waveforms with intermediate PP values. These waveforms will be highly contaminated by off nadir scattering, which causes the low sea ice elevation estimates, and hence negative freeboard when differenced with the local sea level. The authors need to improve their treatment of the Envisat data before it can be considered 'state of the art' and is suitable for publication. (See my specific comments below).

2. Related to this is the waveform interpretation. The authors assert that waveforms with intermediate PP values originate from thin level ice, however these waveforms are conventionally interpreted as showing 'mixed' scattering behavior. The 'conventional' interpretation is backed up by publications which compare altimeter returns with coincident imagery [e.g., Peacock & Laxon (2004), Armitage & Davison (2014)]. As C2 well as this, it is known that sea ice is rarely homogeneous at the scale of altimeter footprints (even SAR footprints), so you would almost always expect mixed scattering behavior to be present in echoes over sea ice. I believe that the waveforms presented in Figure 4 also show mixed scattering behavior – they all have a diffuse scattering component corresponding to the sea ice, and each one has a specular part superimposed on top, presumably corresponding to leads or thin, freshly formed ice. You should plot the absolute power of the waveforms – is the diffuse scattering? I would require much more convincing, including detailed comparison with imagery, and possibly scatterometry (to show roughness), to be convinced by the interpretation that the intermediate waveforms correspond to thin, level ice.

As you and the other reviewers agree on the fact that it is essential to filter radar observations characterized by an intermediate PP value, we now filter these ambiguous data. The threshold values were selected by combining optical imagery with radar observations as you recommended it. As a matter of fact, the freeboard is slightly improved but is still highly negative.

Let's consider your interpretation. If off-Nadir reflections are indeed the cause for the negative freeboard estimates, then the Envisat radar freeboard should be further negative in regions with a high concentration of leads (regions with a high PP). However, it is precisely in these regions that the freeboard is the least underestimated (relatively to CryoSat-2). Thus, this interpretation doesn't really get along with the results we show.

To present the problem in a different way: you recently posted a paper in TC concerning sea-level estimates. To obtain these sea-level estimates, you use one retracking algorithm for sea-ice leads

and one retracker algorithm for open ocean surfaces. But if you estimated the sea level with only one retracker, you would obtain an average sea level elevation in leads 20-30 cm above the elevation you obtained over open ocean surfaces.

For sea level studies, this approach seems fairly resonable as you have 2 very distinct types of surfaces (leads and open ocean). But if you now consider sea ice, there is a wide range of ice types that all have a different impact on the freeboard retrieval. It is the main purpose of our study to describe and correct this phenomenon. Hopefully the new organization of the paper and the new details we provide will help to clarify this.

3. The reference to "ice surface diffusion" and "surface diffusion variability" throughout the manuscript is confusing, and I do not know what the authors are actually referring to. I don't think I have come across this terminology in any other publications on satellite altimetry. You need to clarify, or adopt more conventional terminology. In some parts, it seems that you are implying that the different footprint shape/size changes the surface/volume scattering components of the ice (e.g., page 3, line 14-16). As far as I am aware, the surface/volume scattering depends on the frequency, the angle of incidence, and surface properties like grain size and water/salt content. I don't see how footprint size or shape can affect these properties?

It is a phenomenon widely described in oceanography and it applies even more over sea-ice. In the study by Chelton et al., 2001 (http://geodesy.geology.ohio-state.edu/course/refpapers/Chelton altimeter 02.PDF) it is explained how the footprint size can be impacted by surface roughness (have a look at figure 7b). I believe that, one should say "effective footprint" rather than "footprint" alone to avoid any confusion, which is not commonly done in the literature...

4. I think it should be made clear throughout the manuscript that you are actually comparing the "radar freeboard" rather than "sea ice freeboard" e.g., page 1, line 5. This is particularly important when you're comparing the two instruments. For example, you say that the Envisat freeboard decreases during the season whilst CS2 increases – in actual fact the freeboard is independent of the altimeter (it is a geophysical quantity), but the radar freeboard that is retrieved by the altimeter can be different with different instruments. This distinction has been made in other publications (e.g., Ricker et al, 2014, Armitage and Ridout (2015)) and accounts for the fact that the altimeter freeC3 board may not correspond directly to the ice-snow interface.

You are absolutely right, talking about radar freeboard would avoid lots of confusions. Changes have been operated throughout the manuscript.

5. Finally, I would consider splitting this paper into two. The first would concern the technical aspects of making a consistent sea ice thickness time series from two different altimeters, and evaluation of the data against in situ and airborne data. The second would use the decade+ long time series to do some science! The scientific value of this dataset is large, and it is wasted here – section 3.5 is just two paragraphs. If you retain the 'scientific' part of this manuscript, you should provide some interpretation – what is driving the inter-annual and long term changes of ice thickness? You should also provide maps of the sea ice thickness through the period, for example autumn (Oct&Nov) and spring (Feb&Mar) average thickness.

Here as well the other reviewers share your opinion. We therefore remove this section that will be further developed in a future study.

**Specific comments:**

Throughout the manuscript: the authors consistently refer to "freeboard height" - it is a personal preference but I think that you just need to say "freeboard", and not "freeboard height".

Thanks for the advice. We operated the corresponding changes throughout the manuscript and it does indeed make the manuscript clearer.

**Page 1, line 3: "..free of instrumental error as possible". This is a rather trivial statement (of course you wish to minimize instrumental error) however it also misses the point that sea ice thickness uncertainty is dominated by snow loading error, not instrumental error.** This section was rephrased.

**Page 1, line 4: It's more accurate to say that you compared freeboard during the 2010/11 and 2011/12 sea ice growth seasons.** OK

Page 1, line 10-12: It isn't valid to present a comparison of the EnvisatPP data with CS2 as a significant result because you are using CS2 to calibrate the EnvisatPP data – so the 'improvement' is by construction! The BGEP comparison is more significant.

We agree with you. The message we want to bring here is that the PP correction works for every months and during the 2 ice growth seasons. This part has been rephrased to make the message clearer.

Page 1, line 18-19. It would be interesting to test exactly how much ice volume Envisat is missing in the 'pole hole', by comparison with CS2 and ICESat. The 'circumpolar' claim (here and elsewhere in the manuscript) is arguable, due to the size of the Envisat pole hole.

It is indeed something we would like to do in the future study to emphasis or not the ice thickness estimates bellow 81.5°N.

**Page 2, line 9: "For \*more\* than a decade,..." or "Since 2003,. . ."** OK

Page 2, line 14 and page 3, line 3-19: "LRM" – you should refer to the Envisat data as "pulselimited" rather than "LRM". Low resolution mode is specific to CS2 and is just conventional pulse limited operation.

We now use pulse-limited instead of LRM.

Page 2, line 22: Some references are missing: Ricker et al. (2014), Kurtz et al. (2014), Tilling et al. (2015).

Only Tilling et al. (2015) treats of ice thickness. This reference is now added in this section.

Page 2, line 23-page 3, line 2: The "important question" discussed here is not a question at all: CS2 provides better estimates of ice thickness than Envisat because it was designed to! In the late 90s, the question was asked, how can we improve altimeter design to better capture interannual and seasonal sea ice thickness variability? The answer was CS2 – a SAR altimeter, with very high inclination orbit.

Right, but it is still insightful to understand why C2 is better and we think that it is important to not reduce radar altimeters as an instrumental concept and mission requirement. In fact, it is by FULLY

answering this question that we improve the accuracy of the Envisat freeboard estimates.

**Page 2, line 25-26: The freeboard to thickness conversion uncertainty affects both Envisat and CS2 in the same way, so would not result in a bias in Envisat.**

Yes indeed, this is the message we want to pass through... It is now rephrased. Hopefully in a better way.

**Page 4, line 12: the bandwidth (receive) of SIRAL is the same as Envisat, not similar.** That is correct, thanks.

Page 4 line 27-page 5, line 7: This relates to my major comment above. You need to provide substantial evidence that intermediate PP waveforms "likely result from thin and relatively flat sea ice", as this would be contrary to the current understanding as presented in the literature. You say that filtering these data may bias the sea ice thickness high, however there is no evidence of this in other publications presenting comparisons with in situ data (e.g., Tilling et al (2015)). In fact, including these waveforms produces the extreme negative freeboard maps present in Figure 2b. For me, you would have to develop and demonstrate an extremely robust retracker to make use of intermediate PP Envisat waveforms.

As discussed above, this part has been strongly modified and we now filter ambiguous observations.

**Section 2.3: It is surely not valid to use the exact same processing for Envisat and CS2 (PP thresholds, retracker parameters) given the fundamental difference between the instruments??**

Regarding the PP threshold, there is now a demonstration with the use of collocated images. Regarding the retracker parameters, the issue that we might faced when applying the same retracker to the two senors are now discussed in the new version.

**Page 5, line 10-12: Two different retrackers are used in Laxon et al (2013), hence the need for the bias correction. As a point of reference, the SICCI ATBD is actually based on the CPOM processing presented by Laxon et al (2013).**

You are right. Thanks for pointing at this error.

**Page 5, line 16-19: Has this retracker been demonstrated for Envisat, or just CS2? If not, then you need to do a proper assessment on the Envisat data.**

As Dr. Rinne (RC1) mentions it in his review, the TFMRA retracker have been tested on Envisat by the SICCI group and seems to have good results. Hopefully, Dr. Rinne will provide a reference for this result.

Page 5, line 21-27: Sea level interpolation causes errors because of lack of lead tie points, snagging, or use of a poor geoid/MSS model. Geophysical corrections have a much smaller effect, as I think another reviewer pointed out. Your method for treating sea level interpolation is new and needs to be demonstrated more robustly against current algorithms.

In fact the methodology we use is quite similar as what is generally found in the literature. The main difference is that we do not estimate freeboard height where no tie point is identified. We rephrase this section in order to clarify our methodology.

**Page 6, line 4-6: I believe it was Laxon et al. (2013) who first used the "Warren/50% on FYI" methodology, not Kwok & Cunningham (2015).**

Yes indeed. However, in Laxon et al. (2013) the authors use a binary parametrization (0.5 or 1). What we use in our study is a progressive parametrization (from 0.5 to 1), which was first developed in the study by Kwok & Cunningham (2015).

**Page 6/Figure 1: monthly snow depth – wouldn't it be better to use daily ice type masks and match to individual altimeter orbits? The location/size of the MYI area can vary quite a lot over the course of a month.**

This is indeed a good idea that we will most likely develop in the future. As our study is no longer related to climatological studies, we stick to what is done in the literature.

**Page 7, line 17, Figure 2c: You should introduce figure 2c here or move it – perhaps move it to Figure 3.**

This was also proposed by reviewer #2. We modified the organization of the manuscript so the problem you rise is no longer an issue.

Figure 3: I find the colourbar used for Figure 3 misleading – normally the red-blue "polar" colourbar is centred on zero, to show positive/negative values. It also makes it appear as though the PP is zero in large areas.

We changed back to regular "jet" colormap.

**Page 8, line 4-5: Here is an example of misleading use of "thicker freeboard". The radar freeboards are different, the ice freeboard stays the same.** OK

**Section 3.2: This section will need considerable revision based on my major comments. Page 9, lines 11-18: Is the first part of this paragraph necessary? Consider cutting.** This section was highly modified.

Section 3.3 is good, the most interesting/important development of the paper.

Page 10, line 4-5, Figure 18a,b,j,k. I think it's worth noting that the CS2/EnvisatPP are so similar \*by construction\*. Currently the paper makes is appear like the agreement between EnvisatPP and CS2 is a significant result in itself, but it is simply a consequence of levelling the CS2 against the Envisat data. This doesn't detract from low RMSE or the good agreement seen with the BGEP moorings, but is an important point.

As explained above, we now emphasis more on the good correlation for each month of the period of study than on the general agreement, which is indeed not an actual evidence.

**Section 3.4: I wonder if you could do your evaluation with any other datasets? E.g., Fram Strait moorings have been in place for a long time, Operation IceBridge goes back to 2009, EM-bird data.**

For our next study (more climatological this time), we consider using new validation datasets such as the one you cited. I doubt that measurements obtained by the Fram Strait moorings will match perfectly with the altimetric estimates considering the sea-ice dynamics in this region. Still, it is still worth to give it a try.

Section 3.5: I think this section should be greatly expanded, or else written up as a separate paper. What is driving interannual to decadal thickness variability? This can be done by comparison with ice drift, temperature records, climate indices (e.g., AO). You should compare the Envisat thickness with ICESat. You should present seasonal maps of ice thickness for the entire time period. Are changes in basin mean thickness reflected in changes in volume? What are the implication for heat/freshwater storage?

As mentioned above, this section has been removed and will be part of a full study.

**page 11, line 23: The references should be in chronological** OK

**III-Summary of changes #1:**

With respect to the new version manuscript order:

 $\rightarrow$  The abstract and introduction have been slightly re-written to clearly express the aim of this study and the key steps.

 $\rightarrow$  The freeboard processing is now more detailed (sea level, TFMRA retracker, etc). In addition, we add a comparison with Landsat images to validate the use of our PP thresholds.

 $\rightarrow$  Changes in the freeboard processing chains were applied, all freeboard estimates were recalculated and figures were updated.

→ The ice density parametrization has been modified and is now more in phase with the literature (882 kg/m3 for MYI and 917 kg/m3).

 $\rightarrow$  A short analysis of CryoSat-2 and Envisat waveforms is now provided (sect 3.1)

 $\rightarrow$  An analysis of along-track radar freeboard is now provided (section 3.2).

 $\rightarrow$  Section 3.3 and 3.4 have been inverted.

 $\rightarrow$  The section showing ice thickness time series has been removed and will be part of a future study.

→ Tables with statistical parameters were improved

 $\rightarrow$  In general, the physical impact of ice surface properties on the radar signal is more clearly explained.

---

## Referee Report (RR1)

**Response to the Revision of "Comparison of CryoSat-2 and ENVISAT radar freeboard over Arctic sea-ice: Toward and improved Envisat freeboard retrieval", by Geurreiro et al.**

I believe the revised manuscript is much improved, and I appreciate the authors' efforts to respond to the points brought up by the reviewers.

Regarding the interpretation of the freeboard differences as being due to different responses to surface roughness. I am familiar with this phenomenon, of course, however I hadn't heard of it referred to as 'surface diffusion' in the cryospheric altimetry literature. I do appreciate your point better in the revised manuscript, as the arguments are much better presented. If we forget for a moment the static offset due to only using the one retracker, and assume the error is in the Envisat data, do you think the spatial variability of the freeboard difference is driven by an underestimation of the Envisat range over FYI, or an overestimation of the Envisat range over MYI, or a combination of both? Is there any evidence from the literature (e.g., the sea state bias in oceanography studies) to expect the difference to be in this direction? Whilst I understand how biases can arise due to this effect, it is not clear why you might expect the bias to be in this direction, and some discussion might be worthwhile.

Figure 6 – is there a reason that you only show results from the 2010/11 growth season? For completeness, I think you should also include the 2011/12 growth season in the final version.

Figure 4 caption "the November 2010-March 2012 period" – is this figure an average of the two growth seasons, or is this a typo? I think it would be better to show both growth seasons separately.

Page 7, line 1-3, and elsewhere. "High vs. low delta(Fb)." When you say "High delta(Fb) over FYI" you are actually referring to the smallest differences in freeboard. I find this quite confusing, and I think contributed to some misunderstanding in my previous review. Consider changing the use of high/low delta(Fb) – perhaps the smallest/greatest difference in freeboard or something similar.

Do you have a reason for fitting y(PP) to both years of data together, rather than each year separately? If your argument about the effects of surface roughness are correct then you might expect y(PP) to vary from year to year, and inspecting Figure 5, it does seem that there is a difference – the relationship seems more linear in the 2011/12 season. Either way, you should provide more of a justification for fitting both years together, and also a discussion of interannual variability when you extend the Envisat time series using the means values, particularly as the amount of MYI seen by Envisat will be larger in previous years.

Page 11, line 31 – I don't think something can be "quasi-identical", it is either identical or not. Should read "similar mean/modal values".

---

## Author Response (AR3)

**Responce to RC1**

Response to the Revision of "Comparison of CryoSat-2 and ENVISAT radar freeboard over Arctic sea-ice: Toward and improved Envisat freeboard retrieval", by Geurreiro et al.

**General comments:**

**The revised manuscript is much improved during revision. The potential impact of the paper is still big, this being the first RA-2 methodology for FB (and draft) retrieval that shows good agreement with both CS-2 and BGEP measurements. The impact is more than sufficient to justify publication despite the fact that the authors have decided to omit the circumpolar thickness time series. The methods and data sections would benefit from having more details and rigour. However, I have very little comments to the actual results section now since it is good now. Thus I recommend this paper to be published after minor revisions.**

Thanks a lot for the positive comments and advices.

**Detailed comments:**

**P1L15 onwards. Reading this makes me think that the authors have created a PP time-series. They should clearly keep the fact that it is FB they are after written in the text. Rephrase.**

→ Details were added to this sentence to improve the comprehension.

**P3L24-30 The data source is bit surprising here. Is the SGDR at the aviso site different from the SGDR product from ESA ( https://earth.esa.int/web/guest/-/ra-2-sensor-data-record-1471 )? If it is not, most likely better to cite the ESA documentation of the data instead of a partly French language website. If it is different, please explain how. I have doubts because I don't think the ESA SGDR includes the DTU15 but it is added in the CTOH processing which would then be more than just conversion to a netdf. But I may be wrong. Please check and correct if necessary.**

→ Adding geophysical corrections such as DTU15 in indeed done by the CTOH during the netcdf production. This is now mentioned.

**P4L13-14 It would still be good to list exactly which of the corrections are applied to the range. This would help people reproducing the results.**

→ In the new version, we precise that we use the "model" atmospheric corrections. This allows to be consistent between the two missions as there is no radiometer on board C2. Concerning the rest of the corrections, there is only one parameter in the sgdr.

**P4L20-25 I am sure you can find a better way to cite BGEP data. Must be a paper published somewhere.**

→ Yes indeed. The paper by Melling et al. (1995) is now cited.

**P5L29 Based on the evidence in figure 1, you have shown that these are suitable PP thresholds for two leads per mission. Furthermore, it looks very much like the upper lead in the Envisat**

**frame is already detected as a lead when it is well off-nadir. As the PP threshold values are more or less reasonable, I would suggest that the authors just add a mention of the PP (and other lead detection) thresholds other have used and state that theirs are not too different from them.**

→ In the figure you mention, I do not see where Envisat detects leads that do not exist although I agree that the lead is partly covered with ice.
Unfortunately in the studies by Giles et al. (2008) and Schwegmann et al. (2016), the authors do not mention the threshold they use. We can therefore not compare the PP value between our and theirs studies...

**P5L30 wile → while**

→ done

**P6L6 reflexion → reflection (even though x is not entirely wrong)**

→ done here and everywhere in the manuscript

**P6L16 Please explain the reason why you use a median filter. 2.4 Freeboard retrievals still does not mention filtering out bad echoes at all. I assume the authors trust that the robust TFMRA retracker and the PP thresholds for surface classification will take care of them. This is fair enough, but it should be clearly stated in the manuscript. Same goes for filtering of freeboard values – do you make any kind of reality check to remove unrealistic freeboards due to for example bad correction values? If not, you should mention it.**

→ That is right, we forgot to mention that we filter freeboard values below -1 m and above 2 m as in Schwegman et al. (2016). This is now mentioned in the manuscript.

**P11L23: Please add the formula from figure 6 to the text. y(PP) = … Figure 3 caption: I think leads = black and floes = gray. Check and correct.**

→ done

Response to the Revision of "Comparison of CryoSat-2 and ENVISAT radar freeboard over Arctic sea-ice: Toward and improved Envisat freeboard retrieval", by Geurreiro et al.

**I believe the revised manuscript is much improved, and I appreciate the authors' efforts to respond to the points brought up by the reviewers. Regarding the interpretation of the freeboard differences as being due to different responses to surface roughness. I am familiar with this phenomenon, of course, however I hadn't heard of it referred to as 'surface diffusion' in the cryospheric altimetry literature. I do appreciate your point better in the revised manuscript, as the arguments are much better presented.**
**If we forget for a moment the static offset due to only using the one retracker, and assume the error is in the Envisat data, do you think the spatial variability of the freeboard difference is driven by an underestimation of the Envisat range over FYI, or an overestimation of the Envisat range over MYI, or a combination of both?**

→ That is an excellent question... Unlike for ocean altimetry, glaciologist seek to measure the surface height at Nadir position (over leads or floes). Hence, the larger the footprint, the less accurate the range measurement is. As the "effective radar footprint" is larger over rough surfaces, we might say that the range measurement is more underestimated over MYI than over FYI.

**Is there any evidence from the literature (e.g., the sea state bias in oceanography studies) to expect the difference to be in this direction? Whilst I understand how biases can arise due to this effect, it is not clear why you might expect the bias to be in this direction, and some discussion might be worthwhile.**

→ This phenomenon is indeed very well known in oceanography studies and it is deeply discussed in the book by Chelton et al. (2001). We add the following sentence so the reader know where to find the information.

"In particular, the LE widening is likely to drive an underestimation of the surface level position over rough surfaces (Chelton et al. 2001)."

**Figure 6 – is there a reason that you only show results from the 2010/11 growth season? For completeness, I think you should also include the 2011/12 growth season in the final version.**

→ There is no particular reason except the size of the figure that is quite large. The best is perhaps to put this figure in the supplement material so the curious reader can take a look at them.

**Figure 4 caption "the November 2010-March 2012 period" – is this figure an average of the two growth seasons, or is this a typo? I think it would be better to show both growth seasons separately.**

→ It is indeed a typo; the figure only displays the 2010-11 winter season. The corresponding figure for the 2011-12 season is also added in the supplement material.

**Page 7, line 1-3, and elsewhere. "High vs. low delta(Fb)." When you say "High delta(Fb) over FYI" you are actually referring to the smallest differences in freeboard. I find this quite confusing, and I think contributed to some misunderstanding in my previous review. Consider changing the use of high/low delta(Fb) – perhaps the smallest/greatest difference in freeboard or something similar.**

→ Right, we performed the changes you've suggested.

**Do you have a reason for fitting y(PP) to both years of data together, rather than each year separately? If your argument about the effects of surface roughness are correct then you might expect y(PP) to vary from year to year, and inspecting Figure 5, it does seem that there is a difference – the relationship seems more linear in the 2011/12 season.**
**Either way, you should provide more of a justification for fitting both years together, and also a discussion of interannual variability when you extend the Envisat time series using the means values, particularly as the amount of MYI seen by Envisat will be larger in previous years.**

→ I am not sure I understand your comment. What we are showing in this figure is a physical effect. From one year to another, this effect should remained unchanged...
However, we indeed find small differences between the 2 winter seasons which we did no expect. As we mention it in the manuscript, we attribute these differences to freeboard uncertainty due in particular to a too low amount of fb measurements over MYI (low PP). To minimize the impact of the uncertainty on the y(PP) fit, we decided to fit both winter seasons as we have no proof than one might be better than the other one. This latter aspect is now discussed in the manuscript.

**Page 11, line 31 – I don't think something can be "quasi-identical", it is either identical or not. Should read "similar mean/modal values".**

→ quasi-identical was changed to similar

**Response to RC3**

**General comments:**

**The revised paper has improved significantly and certainly provides more information on the processing steps. However, I still think that the paper can be improved, certainly in terms of clarification. Therefore, see my comments below:**

Thank you for your the helpful and positive comments.

**Specific comments:**

**P1L5: Probably somewhere here, I think it must be stated that both CS2 and ES have been processed using the same algorithm.**

→ This information is now provided in the abstract

**P1L5-L14: For the Abstract, this paragraph is to difficult to follow from my point of view. I would suggest to combine the part about the along track analysis with the part about the gridded freeboard below. May be you can just start like: "The analysis of along track data and gridded radar freeboard in conjunction with Envisat pulse peakiness maps suggests …"**

→ Thank you for your comment. The abstract has been shorten and should be now clearer.

**P1L9-L10: I would avoid parenthesizing here and in general in the Abstract, except for values and numbers.**

→ Only numbers and date are now parenthesized.

**P2L24: I still find the usage of the term "accuracy" somewhat confusing here. I think the higher accuracy in the CS2 measurements mostly is a result of the SAR processing. On the other hand, the unrealistic ES freeboard can be corrected by tuning the retracking algorithm and/or the retracking threshold. This would accommodate the different footprints. And this is what I would call a "bias" correction. But in the end, ES freeboard will still be more noisy than CS2 freeboard, due to CS2 SAR processing. This is what I would associate with "accuracy".**

→ This sentence was slightly rephrased to avoid the use of "accuracy" or any other inappropriate word.

**P2L25: Here and at other places: I am not so sure about the usage of "better". Could "more realistic" provide a better description?**

→ Here "accurate" was replaced by "more realistic".

**P3l32: What does the CTOH netcdfs contain? geo-located waveforms? l1b elevations? What kind of data are you using? Please, be more specific here.**

→ More details are now provided

-Even though more details are now provided in the new version, I am not sure what you mean by "what kind of data are you using" considering the information we already provide. Is it

**better now? If not, could you be more specific on you expectations please?**

**--Yes, I think it is better now. The description is much more detailed now and this makes it easier for the reader to follow.**

**P4L25: Here, for example, "accuracy" is used as I would expect.**

→ Ok

**P9L9: drive(s)**

→ "drive" was replaced by "drives"

**Figure6: Please, add a legend with Envisat (blue) and CryoSat-2 (black) directly in d). Also, in the PDFs, it seems that Envisat= Envisat/PP? Please, check this.**

That is correct, it is Envisat/PP, not Envisat. Thank you for pointing that out. The legend is now added in the figure (here and in the supplement material) and the caption is corrected.

**Table1: Units are missing.**
**Table2: Units are missing.**

→ Units (centimeters) are now provided in both tables.

**Response to the editor**

**Dear authors,**
**thank you for the revisions of the manuscript and consideration of the reviewer's comments. Unfortunately it seems that you have not considered the comments of Reviewer 1. Maybe they were not easy to find because the online system had some troubles after the unintentional withdrawal of your manuscript at some point. Therefore I attach them here, see below. Please respond to these comments and include revisions as appropriate. Sorry for the inconvenience.**

→ Indeed, we did not see these comments on the website. Thank you for pointing that out. We took these new comments in consideration and added the response in this document.

**I can see that that reviewer also had some important remarks which I wanted to make myself. In addition, please add labels a-d to figure 3, and verify that the grey and black dots are correctly described in the caption. I think they are swapped.**

→ They were swapped indeed. Thank you. Changes were made.

**Also, please remove or define diffusivity. You are still using the term in one or two places although it is unusual and undefined.**

→ Right, "diffuse" is usually not employed to characterize a surface. Though, it is commonly used in the literature to refer to the radar signal back-scattering (see Laxon et al. (2013) page 733; Zygmuntovska et al. (2013) page 1417; Ricker et al. (2014) page 1609; etc).
Hence, we replaced "diffuse" by rough when used to characterize a surface and we kept using the term when talking about the radar signal. We hope this solution is fine to you.

**I agree with the reviewer below that the abstract should be improved.**

→ The abstract was shorten and clarified.

**Thank you for your consideration and best regards**

Thanks a lot for your help and constructive comments.

[revised manuscript text omitted]